# Shielded Diffusion: Generating Novel and Diverse Images using Sparse Repellency

**Michael Kirchhof** [1] [2]   **James Thornton** [1]   **Louis Béthune** [1]   **Pierre Ablin** [1]   **Eugene Ndiaye** [1]   **Marco Cuturi** [1]

## Abstract

The adoption of text-to-image diffusion models raises concerns over reliability, drawing scrutiny under the lens of various metrics like calibration, fairness, or compute efficiency. We focus in this work on two issues that arise when deploying these models: a lack of diversity when prompting images, and a tendency to recreate images from the training set. To solve both problems, we propose a method that coaxes the sampled trajectories of pretrained diffusion models to land on images that fall *outside* of a reference set. We achieve this by adding *repellency* terms to the diffusion SDE throughout the generation trajectory, which are triggered whenever the path is expected to land too closely to an image in the *shielded* reference set. Our method is *sparse* in the sense that these repellency terms are zero and inactive most of the time, and even more so towards the end of the generation trajectory. Our method, named **SPELL** for *sparse repellency*, can be used either with a static reference set that contains protected images, or dynamically, by updating the set at each timestep with the expected images concurrently generated within a batch, and with the images of previously generated batches. We show that adding SPELL to popular diffusion models improves their diversity while impacting their FID only marginally, and performs comparatively better than other recent training-free diversity methods. We also demonstrate how SPELL can ensure a shielded generation away from a very large set of protected images by considering all 1.2M images from ImageNet as the protected set.

[1]Apple [2]University of Tübingen. Correspondence to: Michael Kirchhof <Contact info see website>.

*Proceedings of the $42^{nd}$ International Conference on Machine Learning*, Vancouver, Canada. PMLR 267, 2025. Copyright 2025 by the author(s).

## 1. Introduction

Diffusion models (Song et al., 2021; Ho et al., 2020) are by now widely used for engineering and scientific tasks, to generate realistic signals (Esser et al., 2024) or structured data (Jo et al., 2022; Chamberlain et al., 2021). Diffusion models build upon a strong theoretical foundation used to guide parameter tuning (Kingma & Gao, 2023) and network architectures (Rombach et al., 2022), and are widely adopted thanks to cutting-edge open-source implementations. As these models gain applicability to a wide range of problems, their deployment reveals important challenges. In the specific area of text-to-image diffusion (Nichol et al., 2022; Saharia et al., 2022), these challenges can range from an expensive compute budget (Salimans & Ho, 2022) to a lack of diversity (Ho & Salimans, 2022; Shipard et al., 2023) and/or fairness (Cho et al., 2023; Shen et al., 2024).

**Controllable Generation.** We focus on the problem of ensuring that images obtained from a model are sufficiently different from a reference set. This covers two important use-cases: *(i)* the purveyor of the model wants images generated with its model to fall *outside* of a reference set of protected images; *(ii)* the end-user wants high diversity when generating multiple images with the same prompt, in which case the reference set could consist of all previously generated images, or even other images generated concurrently in a batch. While the problem of avoiding generating images in a protected training set (Carlini et al., 2023) originates naturally when deploying products, that of achieving diversity within a batch of generated images with the same prompt should not arise, in theory, if diffusion models were perfectly trained. However, as shown for instance by Sadat et al. (2024), state-of-the-art models that incorporate classifier-free guidance (Ho & Salimans, 2022, CFG) do a very good job at outputting a first picture when provided with a prompt, but will typically resort to slight variations of that same image when re-prompted multiple times. This phenomenon is illustrated in Figure 1 for three popular diffusion models, Stable Diffusion 3 (Esser et al., 2024), Simple Diffusion (Hoogeboom et al., 2023) and MDTv2 (Gao et al., 2023).

**Contributions.** We propose a guidance mechanism coined *sparse repellency* (SPELL), which repels the backward diffusion at generation time away from a reference set of images.

**Stable Diffusion 3**
*Prompt: "A close-up of an apple"*

**Simple Diffusion**
*Prompt: "The Eiffel Tower"*

**MDTv2**
*ImageNet class 145 (king penguin)*

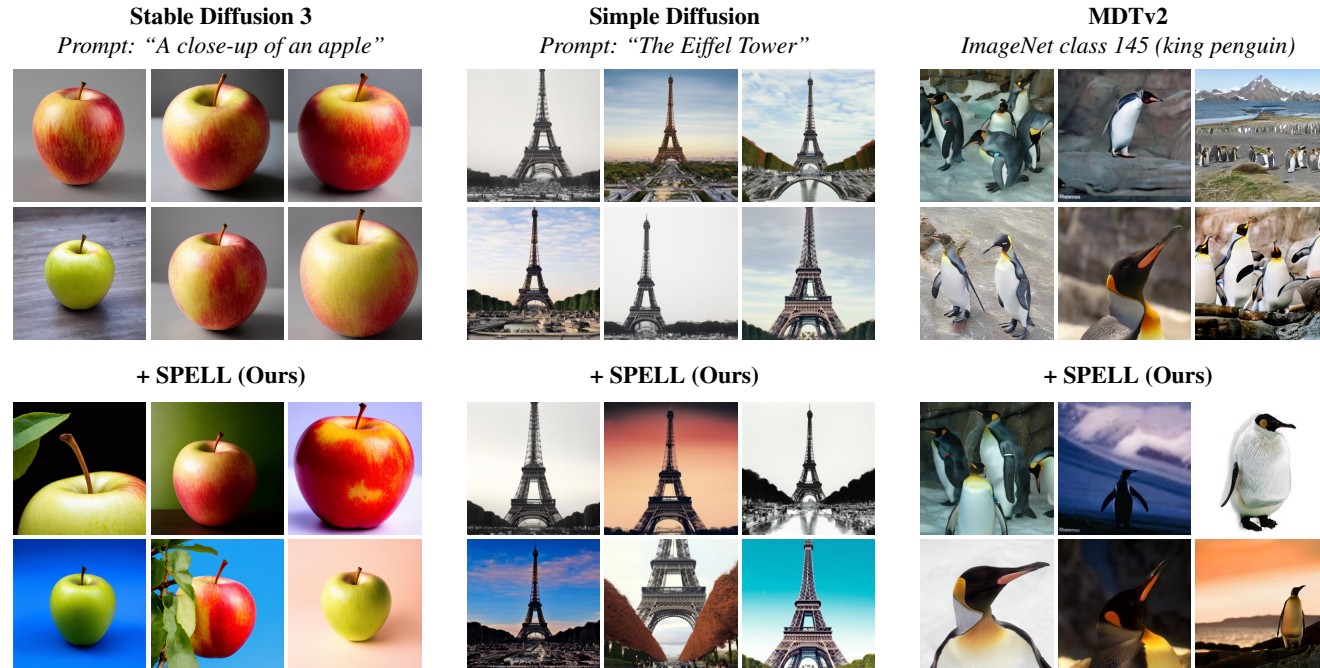

**+ SPELL (Ours)**          **+ SPELL (Ours)**          **+ SPELL (Ours)**

*Figure 1.* SPELL interventions can change the diffusion trajectory of any pre-trained diffusion model by self-avoiding other images generated, in the same or previous batches (and also any other non-generated image). This makes SPELL achieve a higher diversity above, with prompts, and noise seeds as the base models. We provide more qualitative examples in Appendix N.

- SPELL interventions are sparse by design; they only consider very few active shielded images (typically one) at each time $t$, and happen mostly early in the generation.

- SPELL can increase the *diversity* of outputs by dynamically updating the shielded reference set to be images that were generated in previous batches and those that are *expected* to be generated in the current batch.

- We apply SPELL to numerous state-of-the-art diffusion models and find that the generated images better reflect the diversity of the true data (see Figure 1) with a marginal or even no increase in the Fréchet inception distance (FID).

- SPELL is simply parameterized by $r$, the shields' radius. We show that increasing $r$ increases accordingly the output's diversity, with a better diversity-precision trade-off than other recently proposed methods (Sadat et al., 2024; Corso et al., 2024; Kynkäänniemi et al., 2024).

- We scale SPELL to a reference set of millions of images. This enables a second use-case: We shield the whole ImageNet-1k train dataset and generate images that are *novel*, without requiring to filter or regenerate images.

## 2. Background

**Diffusion Models,** also known as score-based generative models (Song et al., 2021; Song & Ermon, 2019; Ho et al., 2020), enable sampling from data distribution $p_{\text{data}}$ on sup-

port $\mathcal{X} \subset \mathbb{R}^d$, such as an image dataset, by simulating the reverse stochastic differential equation (SDE) (Haussmann & Pardoux, 1986; Anderson, 1965), initialised from some easy to sample prior $p_1 \in \mathcal{P}(\mathbb{R}^d)$, $\mathbf{X}_1 \sim p_1$:

$$\mathrm{d}\mathbf{X}_t = [f(t, \mathbf{X}_t) - g^2(t)\nabla \log p_t(\mathbf{X}_t)]\mathrm{d}t + g(t)\mathrm{d}B_t, \quad (1)$$

where $(B_t)_t$ denotes Brownian motion and $p_t$ is defined as the density of $\mathbf{X}_t$ from forward process:

$$\mathrm{d}\mathbf{X}_t = f(t, \mathbf{X}_t)\mathrm{d}t + g(t)\mathrm{d}B_t \quad \mathbf{X}_0 \sim p_0 := p_{\text{data}}, \quad (2)$$

for drift $f : [0, 1] \times \mathcal{X} \to \mathcal{X}$ and diffusion scale $g : [0, 1] \to \mathbb{R}$, where the time $t$ is increasing in equation 2 and time decreasing in equation 1. The score term $\nabla \log p_t(\mathbf{X}_t)$ is typically approximated by a neural network through denoising score matching (Vincent, 2011).

**Training.** The solution to forward diffusion in equation 2 for an affine drift is of the form $\mathbf{X}_t = \alpha_t \mathbf{X}_0 + \sigma_t \varepsilon$, where $\varepsilon \sim \mathcal{N}(\mathbf{0}, \mathbb{I})$ for some coefficients $\alpha_t \in \mathbb{R}$, $\sigma_t \in \mathbb{R}$ (Song et al., 2021; Särkkä & Solin, 2019). The intractable score term may be expressed via denoiser using Tweedie's formula (Efron, 2011; Robbins, 1956): $\nabla \log p_t(x_t) = \frac{\alpha_t \mathbb{E}[\mathbf{X}_0 | \mathbf{X}_t = x_t] - x_t}{\sigma_t^2}$. Hence rather than estimating the score directly, one may approximate $\mathbb{E}[\mathbf{X}_0 | \mathbf{X}_t = x_t]$ via regression, by minimizing:

$$\theta^\star := \arg\min_\theta \mathbb{E}_{\mathbf{X}_t, \mathbf{X}_0} \|D_\theta(t, \mathbf{X}_t, y) - \mathbf{X}_0\|^2 \quad (3)$$

known as *mean*-prediction, for optional condition denoted $y$, then estimate the score via $\nabla \log p_t(x_t \mid y) \approx s_{\theta^\star}(t, x_t, y) := (\alpha_t D_{\theta^\star}(t, x_t, y) - x_t)/\sigma_t^2$. Notice that we do not train model parameters in this work, and will always assume that $\theta^\star$ is given by the purveyor of a model.

**Conditioning and Guidance.** Conditional diffusion models requires access to the conditional score $\nabla \log p_t(x_t \mid y)$ for some condition $y$ such as text or label. It is typically approximated either with explicit conditioning during training of the score / denoising network using equation 3 or as post-hoc additional guidance term added to the score. Given diffusion models have lengthy training procedures, likely due to their high variance loss (Jeha et al., 2024), it is desirable to guide diffusion models with inexpensive post-training guidance (Dhariwal & Nichol, 2021; Zhang et al., 2023; Denker et al., 2024), using e.g. classifier guidance (Dhariwal & Nichol, 2021)

$$\nabla \log p_t(x_t \mid y) = \nabla \log p_t(x_t) + \gamma \nabla \log p_t(y \mid x_t) \quad (4)$$

whereby the gradient $\nabla \log p(y \mid x_t)$ of classifier $p(y \mid x_t)$ for label $y$ is added to the score, heuristically multiplied by a scalar $\gamma \geq 1$ for increased guidance strength. Another approach which circumvents training a time-indexed classifier is using the approximation $p(y \mid x_t) \approx p(y|\mathbf{X}_0 = D_{\theta^\star}(t, x_t))$, for a pretrained denoiser $D_{\theta^\star}$ alias Diffusion Posterior Sampling (DPS) (Chung et al., 2023).

**Classifier-Free Guidance and Lack of Diversity in text-to-image Diffusion Models.** Classifier-free guidance (CFG) (Ho & Salimans, 2022) is the dominant conditioning mechanism in text-to-image diffusion models, sharing properties with both explicit training and guidance. Similar to classifier guidance, CFG may be used to increase guidance strength but without resorting to approximating density $p(y \mid x_t)$. Instead, the difference $\nabla \log p(y \mid x_t) = \nabla \log p(x_t \mid y) - \nabla \log p_t(x_t)$ is used as a guidance term, where each term is approximated with the same conditional network: $\nabla \log p(x_t \mid y) \approx s_\theta(t, x_t, y)$ and $\nabla \log p(x_t) \approx s_\theta(t, x_t, \emptyset)$, for null condition $\emptyset$, trained as in equation 3. Adding CFG to the unconditional score yields $\nabla \log p_t(x_t \mid y) \approx \gamma s_\theta(t, x_t, y) - (\gamma - 1)s_\theta(t, x_t, \emptyset)$. Despite its widespread popularity and good performance, CFG weighting is heuristic. It is not clear what final distribution is being generated; and practitioners observe a lack of diversity in generated samples (Somepalli et al., 2023; Chang et al., 2023; Wang et al., 2024).

## 3. Sparse Repellency

In this section, we introduce SPELL. We first give a geometrical intuition of how its repellency terms steer the diffusion trajectory out of shielded areas. Then, we make a deeper dive into theoretical connections to DPS (Chung et al., 2023) and particle guidance (Corso et al., 2024).

**Setup.** Our goal is to sample from the data distribution $p_0$ whilst satisfying the important requirement that generated samples $\mathbf{X}_0 \sim p_0$ are far enough away from each element of the reference set of *repellency* images $z_i \in \mathcal{X}, k = 1, \ldots, K$. That set may be populated by real-world protected images, samples generated in earlier batches, images expected to be generated by other trajectories in the current batch, or a mix of all these types. More formally, we wish to sample a conditioned trajectory $\mathbf{X}_t \mid (\mathbf{X}_0 \notin S)$, where $S$ is the collection of *shields*, i.e. balls of radius $r > 0$ around repellency images, $S := (\cup_k B_k)$ with $B_k = \{x \in \mathcal{X} : \|x - z_k\|_2 \leq r\}$. A brute-force mechanism to guarantee generation outside of $S$ is to generate and discard: resample multiple times both initial noise and Brownian samples, follow the diffusion trajectory and repeat until a generated image falls outside of $S$. In the context of computationally expensive diffusion models, this would be wasteful and inefficient. Instead, we seek a mechanism which satisfies the protection in each generation, for any conditional or unconditional diffusion model.

**A Geometric Interpretation of SPELL.** To ensure that generation falls outside of the shielded set $S$, we modify the diffusion trajectory at each time step, as presented in Figure 2, without having to discard any samples. To do so, we correct the trajectory whenever the *expected* output, $\mathbb{E}[\mathbf{X}_0 \mid \mathbf{X}_t = x_t]$ falls within a shield. Here the expected given current state $x_t$, is approximated by the denoising network $D_{\theta^\star}(t, x_t)$. Using the notation $\hat{x}_0 = D_{\theta^\star}(t, x_t)$, we test whether for any index $k$ one has $\|\hat{x}_0 - z_k\|_2 < r$. If that is the case, the minimal modification $\delta$ that can ensure $\|\hat{x}_0 + \delta - z_k\|_2 \geq r$ is

$$\delta_k(\hat{x}_0) := \frac{(\hat{x}_0 - z_k)r}{\|\hat{x}_0 - z_k\|_2} - (\hat{x}_0 - z_k).$$

Across all $k = 1, \ldots, K$, we modify the trajectory only for those $k$ that $\hat{x}_0$ is too close to, giving

$$\Delta = \sum_{k=1}^K \mathbf{1}_{B_k}(\hat{x}_0) \cdot \delta_k(\hat{x}_0) \quad (5)$$

$$= \sum_{k=1}^K \sigma_{\text{relu}}\left(\frac{r}{\|\hat{x}_0 - z_k\|_2} - 1\right) \cdot (\hat{x}_0 - z_k) \in \mathbb{R}^d$$

where the set of indices $k$ that the ReLU is non-zero for at each individual timestep is usually very small. Under the assumption that all of their shields $B_k$ are disjoint, for example when the radius $r$ is small enough, this update strictly ensures that $\hat{x}_0 + \Delta \notin S$. When shields overlap, we do not have such a guarantee. While more complicated projection operators might still yield exact updates in that case, they would involve the resolution of quadratic program. We take the view in this paper that $\Delta$ strikes a good balance between accuracy and simplicity.

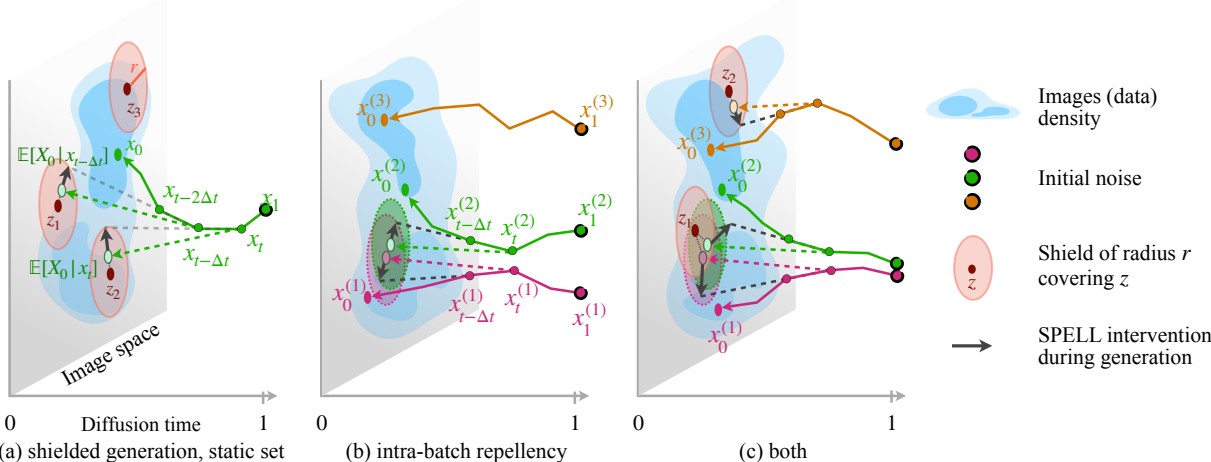

*Figure 2.* (a) At time $t$, by computing $\mathbb{E}[\mathbf{X}_0 \mid \mathbf{X}_t = x_t]$, we detect that the trajectory is headed (in expectation) into the shield of radius $r$ centered around $z_2$. Our sparse repellency (SPELL) term depicted as a black arrow adds a correction when generating $x_{t-\Delta t}$ to ensure that the trajectory is pushed out of the shield. This is again in the case in the next step, when starting from $x_{t-\Delta t}$. (b) In batched generation, the shields are dynamically recreated at every iteration around each trajectory's expected output. This prevents two elements in the batch, $x_t^{(1)}$ and $x_t^{(2)}$, from generating the same output. (c) Both approaches can be combined to yield a diverse set of images that won't fall into protected images and previously or concurrently generated images.

Figure 2(a) visualizes the repellence mechanism away from static protected images while Figure 2(b) shows how it dynamically repels from trajectories within the same batch. The batch generation produces $B$ samples $x_0^{(b)}$ in parallel and its repellency mechanism uses a time-evolving set of repellency points $z_{k,t} = \mathbb{E}[\mathbf{X}_0 \mid \mathbf{X}_t = x_t^{(k)}]$ corresponding to the currently predicted end-state of each sample in the batch. Overall, this leads to a blue-noise like coverage of the distribution, as we visualize in 2D for the two-moons dataset in Figure 3. As we made no further assumptions on $z_k$, these mechanisms can be mixed as in Figure 2(c) to enable diverse generation across arbitrary numbers of batches. This makes it possible to generate large numbers of diverse images even when the VRAM for each batch is limited. Note that SPELL is a post-hoc method that does not require retraining and can be applied to any diffusion score, in RGB space or latent space, unguided or classifier-free guided. Appendix D provides pseudo code and further implementation details.

**SPELL as a DPS guidance term**. We propose to derive SPELL as a guidance mechanism, by Bayes' rule

$$\nabla_{x_t} \log p_t(x_t \mid x_0 \notin S) = \nabla_{x_t} \log p_t(x_t) \\ + \nabla_{x_t} \log p_{0|t}(x_0 \notin S \mid x_t)$$

Hence, we may sample $\mathbf{X}_t \mid x_0 \notin S$ by adjusting a pre-trained score function and simulating:

$$\tilde{s}_t(\mathbf{X}_t, S) = \nabla \log p_t(\mathbf{X}_t) + \nabla \log p_{0|t}(\mathbf{X}_0 \notin S \mid x_t)$$
$$d\mathbf{X}_t = \left[ f(t, \mathbf{X}_t) - g(t)^2 \tilde{s}_t(\mathbf{X}_t, S) \right] dt + g(t) dB_t. \quad (6)$$

The term $\log p_{0|t}(x_0 \notin S \mid x_t)$ in the score adjustment is known as Doob's $h$ transform, and provides a broadly appli-

cable approach to conditioning and guiding diffusions. Unfortunately, Doob's $h$ transform is generally intractable. We may however appeal to diffusion posterior sampling (Chung et al., 2023) and approximate $p_{0|t}$ as a Gaussian with mean $\hat{x}_0 \approx \mathbb{E}[\mathbf{X}_0 \mid x_t]$, which is available from diffusion model pre-training, see Section 2. This approximation results in the following correction:

$$\nabla \log p_{0|t}(\mathbf{X}_0 \notin S \mid x_t) \approx \sum_{k=1}^{K} \omega(\|\hat{x}_0 - z_k\|_2, r) \cdot (\hat{x}_0 - z_k),$$

where $\omega(\cdot, r)$ is a weighting factor detailed in Appendix A that decreases in its first variable. This DPS approximation is similar to SPELL in that both push away trajectories in the directions $(\hat{x}_0 - z_k)$, weighted by a factor that depends on $r$ and the distance $\|\hat{x}_0 - z_k\|_2$. The difference is that DPS based on Gaussians provides a soft guidance that slowly vanishes as $\hat{x}_0$ moves away from $z_k$, and not a hard guarantee that we respect the protection radii around each $z_k$. We have struggled in preliminary experiments to set hyperparameters of such "softer" DPS schemes, because the weight factor to scale the Gaussian by ultimately depends on the magnitude of the likelihood of the shields, which is unknown, and because the Gaussian's weight never becomes exactly zero, hindering sparsity. This is why we focus our attention on the simpler and much cheaper SPELL.

**(Intra-batch) SPELL and Particle Guidance**. When using SPELL to promote diversity within the generation of a single batch (but without the more general protection against arbitrary or previously generated images), SPELL can be re-

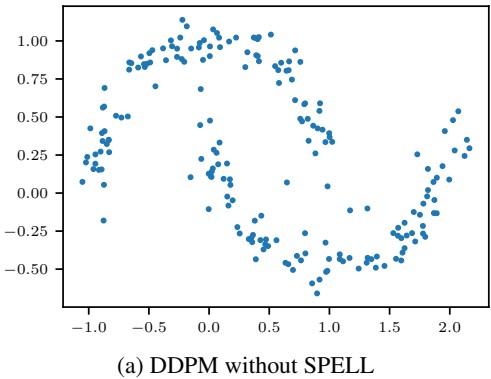

(a) DDPM without SPELL

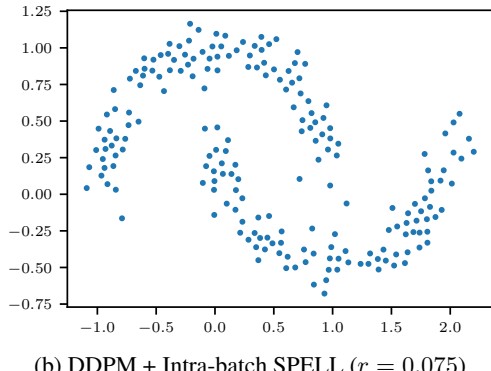

(b) DDPM + Intra-batch SPELL ($r = 0.075$)

*Figure 3.* 200 samples generated with an unconditional DDPM diffusion model on the two moons toy dataset, with and without intra-batch SPELL with minimum shield radius of $r = 0.075$. SPELL's shields lead to a blue-noise-like coverage of the distribution.

lated to the self-interacting particle guidance (PG) approach proposed by (Corso et al., 2024). That approach defines an interacting energy potential $\phi_t$ at time $t$, using the locations in space of all $B$ particles within a batch at time $t$. The gradient of that potential w.r.t. each particle, $\Delta^{(i)} = \nabla_{x_t^{(i)}} \log \Phi_t(x_t^{(1)}, \dots, x_t^{(B)})$ is then used to correct each individual trajectory to guarantee diversity. In contrast to this approach, SPELL draws insight on the expected *future* locations of points, at the end of the trajectory, i.e on the *expected denoised* images $\hat{x}_0^{(i)}$ and $\hat{x}_0^{(k)}$, where $\hat{x}_0 = D_{\theta^\star}(t, x_t)$. Indeed, the correction for each particle is explicitly given as:

$$\Delta^{(i)} := \sum_{k=1}^{K} \sigma_{\text{relu}} \left( \frac{r}{\|\hat{x}_0^{(i)} - \hat{x}_0^{(k)}\|_2} - 1 \right) \cdot (\hat{x}_0^{(i)} - \hat{x}_0^{(k)}). \quad (7)$$

The $B$ correction terms $\Delta^{(i)}$ considered by SPELL cannot be seen to our knowledge as the gradients of an interacting potential. While we prove in Appendix C that $h(x) = \sigma_{\text{relu}}(\frac{r}{\|x\|} - 1)x$ is a conservative field (i.e. the gradient of a potential), we find no guarantee for the more complex SPELL updates above which involve compositions of $h$ with the denoiser $D_{\theta^\star}$. Even if SPELL was a conservative field, the biggest difference between PG and SPELL is that PG defines dense interventions between all particles using soft-vanishing kernels that are never zero and thus always perturb diffusion trajectories. SPELL, conversely, intervenes sparsely and rarely, both in time and w.r.t. points in the reference set. As a result, the original diffusion process is less perturbed, notably towards the end of a trajectory, and SPELL scales to large reference sets of millions of shields.

**Overcompensation**. While our method gives the exact weight required to land outside the shielded areas in Equations (5) and (7), we have experimented with scaling these $\Delta$ terms by an *overcompensation* multiplier $\lambda$. Intuitively, the larger that multiplier, the earlier the trajectory will be lead out of the shielded areas, with the possible downside

of getting more hectic dynamics. We illustrate this addition in Figure 5 with a value $\lambda = 1.6$, which we find to work favorably across multiple models.

# 4. Related Works

Most closely related to our method is that of Corso et al. (2024), who promote diversity through an intra-batch repulsion term, used as guidance for pre-trained diffusion models. Similarly to our work the repulsion term can be applied at $\mathbb{E}[\mathbf{X}_0|x_t]$ or on features. The primary difference is the sparsity of the repulsion term in SPELL, and using this on a fixed reference set in addition to intra-batch.

Chen et al. (2024) is also closely related. Here, the authors apply anti-memorization guidance to pretrained models through three terms: desspecification; caption de-duplication and dissimilarity guidance. Similar to Sadat et al. (2024), desspecification adjusts the CFG scale dependent on nearest neighbor in the training data to $\mathbb{E}[\mathbf{X}_0|x_t]$. Caption de-duplication adds a negative guidance term (Liu et al., 2022) based on the network applied to the caption of the nearest neighbor. Finally, dissimilarity guidance applies an additional guidance term based on similarity between $\mathbb{E}[\mathbf{X}_0|x_t]$ and nearest neighbor search. Our approach is most similar to dissimilarity guidance using multiple nearest neighbors and carefully chosen guidance scale to encourage generated samples to be outside a radius of reference points.

Since initial release of our work, Koulischer et al. (2024) has provided more theoretical grounding for weighting negative guidance (Liu et al., 2022) based on an estimated class-based likelihood. Kim et al. (2025) demonstrates negative guidance using the empirical score of a reference set of images. Unlike aforementioned guidance based approaches, Thornton et al. (2025) introduces a gradient-free SMC-based sampler that encourages diversity by negatively weighting the energy in a Feynman Kac potential, demonstrated only on toy examples.

*Table 1.* SPELL improves the diversity of text-to-image and class-to-image diffusion models considerably, at only a small trade-off in terms of precision. The results are reported as mean $\pm$ std, computed over 5 independent runs with different seeds over the full dataset.

| Model | Setup | Recall ↑ | Vendi ↑ | Coverage ↑ | Precision ↑ | Density ↑ | FID ↓ | FD$_{DINOv2}$ ↓ |
|---|---|---|---|---|---|---|---|---|
| Latent Diffusion | text-to-image | 0.236 ± 0.003 | 2.527 ± 0.005 | 0.447 ± 0.001 | **0.559** ± 0.000 | **0.768** ± 0.002 | **9.501** ± 0.024 | 106.244 ± 0.384 |
| + SPELL (Ours) | text-to-image | **0.289** ± 0.003 | **2.695** ± 0.002 | **0.457** ± 0.001 | 0.551 ± 0.001 | 0.745 ± 0.002 | 9.554 ± 0.043 | **98.761** ± 0.441 |
| SD3-Medium | text-to-image | 0.379 ± 0.004 | 3.749 ± 0.005 | **0.294** ± 0.000 | **0.313** ± 0.001 | **0.345** ± 0.001 | **20.103** ± 0.090 | **230.248** ± 0.812 |
| + SPELL (Ours) | text-to-image | **0.483** ± 0.002 | **4.711** ± 0.013 | 0.229 ± 0.001 | 0.211 ± 0.002 | 0.213 ± 0.002 | 35.174 ± 0.153 | 482.246 ± 0.948 |
| Simple Diffusion | text-to-image | 0.230 ± 0.003 | 2.799 ± 0.006 | 0.355 ± 0.002 | **0.441** ± 0.001 | **0.556** ± 0.002 | **19.879** ± 0.003 | **245.138** ± 0.586 |
| + SPELL (Ours) | text-to-image | **0.248** ± 0.002 | **2.886** ± 0.005 | 0.355 ± 0.001 | 0.433 ± 0.002 | 0.541 ± 0.002 | 19.959 ± 0.033 | 245.748 ± 0.562 |
| EDMv2 | class-to-image | 0.589 ± 0.002 | 11.645 ± 0.022 | **0.551** ± 0.002 | **0.518** ± 0.002 | **1.404** ± 0.005 | **3.377** ± 0.022 | **68.452** ± 0.298 |
| + SPELL (Ours) | class-to-image | **0.600** ± 0.002 | **11.806** ± 0.013 | 0.547 ± 0.001 | 0.508 ± 0.001 | 1.364 ± 0.005 | 3.456 ± 0.021 | 68.909 ± 0.161 |
| SD3-Medium-Class | class-to-image | 0.143 ± 0.002 | 8.861 ± 0.028 | **0.202** ± 0.002 | **0.323** ± 0.002 | **0.801** ± 0.005 | **22.246** ± 0.020 | **328.032** ± 0.571 |
| + SPELL (Ours) | class-to-image | **0.206** ± 0.002 | **12.190** ± 0.032 | 0.146 ± 0.001 | 0.181 ± 0.002 | 0.420 ± 0.006 | 38.709 ± 0.054 | 478.286 ± 0.553 |
| - MDTv2 | class-to-image | 0.623 ± 0.002 | 12.546 ± 0.021 | 0.505 ± 0.001 | 0.401 ± 0.002 | 1.020 ± 0.002 | 4.884 ± 0.052 | 133.175 ± 0.721 |
| + SPELL (Ours) | class-to-image | **0.634** ± 0.002 | **12.772** ± 0.027 | 0.505 ± 0.001 | **0.407** ± 0.001 | **1.029** ± 0.005 | **4.381** ± 0.047 | **122.125** ± 0.291 |

## 5. Experiments

We now show that SPELL increases the diversity of modern text-to-image and class-conditional diffusion models (Section 5.2), with a better trade-off than other recent diversity methods (Section 5.3). We quantify the sparsity of SPELL interventions in Section 5.4. In Section 5.6, we demonstrate SPELL's scalability and a new use-case, shielded generation, by generating novel ImageNet images while shielding all 1.2 million ImageNet-1k train images.

### 5.1. Experimental Setup

In the class-to-image setup, we use Masked Diffusion Transformers (MDTv2) (Gao et al., 2023), EDMv2 (Karras et al., 2024), and Stable Diffusion 3 Medium (SD3) (Esser et al., 2024), three recent state-of-the-art diffusion models. We use the pretrained model checkpoints to generate 50,000 256x256 images of ImageNet-1k classes(Deng et al., 2009) without and with SPELL and compare them to the original ImageNet-1k images. We use the validation dataset as a comparison, since we will conduct experiments that repel from the training dataset in Section 5.6, which would render comparisons to the training dataset meaningless. Given the text-to-image setting, we follow (Esser et al., 2024) using *"a photo of a class_name"* as caption.

In our text-to-image setup, we use SD3, Latent Diffusion (Rombach et al., 2022), and RGB-space Simple Diffusion (Hoogeboom et al., 2023) in resolution 256x256. For the latter two, we use the checkpoints of Gu et al. (2023). Details on hyperparameters are provided in Appendix D. We evaluate these models on CC12M (Changpinyo et al., 2021), a dataset of *(caption, image)* pairs, with captions ranging between 15 and 491 characters. As we aim to investigate whether diffusion models with and without SPELL capture the full diversity of images related to each prompt, we create a sub-dataset of captions with multiple corresponding images, which gives a ground-truth target diversity. This gives a one-to-many setup with 5000 captions and 4 to 128

images each (in total 41,596 images). We explain the construction of this dataset in Appendix E.

To evaluate diversity, we track the recall (Kynkäänniemi et al., 2019), coverage (Naeem et al., 2020), and Vendi score (Friedman & Dieng, 2023). To evaluate image quality, we use precision (Kynkäänniemi et al., 2019) and density (Naeem et al., 2020). We track these metrics per class/prompt and average across classes/prompts. To measure whether the generated images match the true image distributions, we use the marginal FID (Heusel et al., 2017) and the marginal Fréchet Distance with DINOv2 features (FD$_{DINOv2}$, Stein et al. (2024); Oquab et al. (2024)).

### 5.2. Benchmark

We first examine whether adding SPELL post-hoc increases the diversity of trained diffusion models. To this end, we compare each diffusion model to the same model run with the same random generation seeds but with SPELL. In particular, we use intra-batch repellency together with repellency from previously generated batches, to enable repellency across the up to 128 images per prompt/class.

Table 1 shows that SPELL consistently increases the diversity, both in terms of recall and Vendi score, across all text-to-image and class-to-image diffusion models. This demonstrates that SPELL works independent of the model architecture and the space the models diffuse in (RGB space for Simple Diffusion, VAE space for all others). Coverage remains within -1% to +2% of its original value in all models except SD3. The difference between coverage and recall is that coverage uses a more tight neigborhood radius to determine whether an image of the original dataset is covered by the generated ones. In other words, coverage measures a form of dataset match, which can counter-intuitively be decreased by more diverse outputs if the diversity takes different forms or is higher than in the reference dataset. This stands out most for SD3, which was not trained on the reference datasets ImageNet-1k/CC12M. We find that SPELL

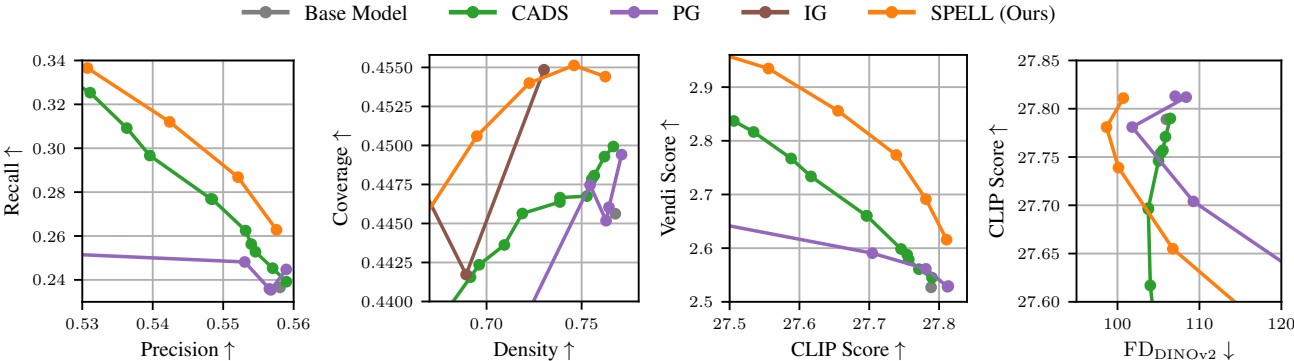

*Figure 4.* **Latent Diffusion on CC12M**. The three plots on the left highlight how the hyperparameters of diversity methods trade off image quality (x-axes) and diversity metrics (y-axes). SPELL provides a better trade-off than other concurrent approaches. In the rightmost plost, highlighting 2 quality metrics, SPELL also shines. IG is not visible on all plots as it strongly decreases image quality.

correctly helps SD3 generate images that are generally more diverse, as evidenced by the 26% and 37% increases in the reference-dataset-free Vendi score, but in other attributes than in the reference datasets, explaining the decrease in coverage. Out of the six experiments, precision and density decrease very slightly in three of them, increase for one, and decrease more clearly in 2 (when using SD3). This trade-off between diversity and precision is common in the literature (Kynkäänniemi et al., 2024; Sadat et al., 2024; Corso et al., 2024), and we show in the next section that SPELL provides more favorable Pareto fronts than alternative recent methods. This tradeoff improves the overall $FD_{DINOv2}$ score considerably in Latent Diffusion and MDTv2, while staying within 3% of the original value on Simple Diffusion and EDMv2, and increasing on SD3. Overall, we find that SPELL increases the diversity considerably across all models, with only minor tradeoffs in precision.

### 5.3. Comparison to Other Diversity-inducing Methods

We now consider SPELL's hyperparameter, the repellency radius $r$. Intuitively, a small radius will only prevent generating a nearly-same image twice, which increases the diversity without compromising on the closeness to the train dataset (precision/density). Choosing to further increase the repellency radius $r$ should add more diversity while trading-off precision, as SPELL pushes the repellency trajectories to explore novel modes further outside the train distribution. We find in Figure 26 of Appendix M that this intended effect plays out in practice. We also find that there is a sweet-spot for $r$ where the diversified samples better reflect the true image distribution than the base model, enhancing $FD_{DINOv2}$ and diversity without decreasing precision.

There are some recent methods enable controlling similar diversity-precision trade-offs. Namely, Interval Guidance (Kynkäänniemi et al., 2024, IG) applies CFG in a limited time interval in the middle of the backward diffu-

sion. Condition-annealed diffusion sampling (Sadat et al., 2024, CADS) noises the text or class condition that guides the CFG, lowering the noise in later timesteps. Closer in spirit to our proposal, particle guidance (Corso et al., 2024) adds a gradient potential to the backward diffusion at every timestep, such that the intra-batch diversity is increased. We reimplement and tune these baselines (Appendix F) to compare them against SPELL.

Figure 4 shows that SPELL achieves more favorable trade-off curves in three different diversity vs quality Pareto fronts (recall vs precision, coverage vs density, and Vendi score vs CLIP Score) as well as in a prompt-adherence vs quality Pareto front (CLIP vs $FD_{DINOv2}$). At low repellency radii $r$, the diversity is achieved without reducing the CLIP score. In Appendix K, we find that SPELL generates diverse images both for short and for longer, more specific prompts.

One reason for SPELL's improved performance is sparsity. While other methods change the diffusion trajectories at every timestep and every image, due to it's ReLU weighting, guidance is not applied if the diffusion trajectories are already heading to a diverse outputs. This leads to increased diversity while leaving high-precision images unchanged. We study this sparsity further in the following section.

### 5.4. Sparsity Analysis

This section investigates the dynamics of when and how strongly SPELL's corrective interventions arise. Figure 5a tracks the magnitude of the SPELL correction vector $\|\Delta\|_2$ normalized by that of the diffusion score vector $\|\nabla \log p_t(x_t|y)\|_2$. We track this relative magnitude throughout 452 backward trajectories for 50 prompts of CC12M with both intra- and inter-batch repellency (Equations (5) and (7)). Appendix H adds further setups with similar results. We find that the repellency correction is typically small. Its magnitude is most often less than 5% of

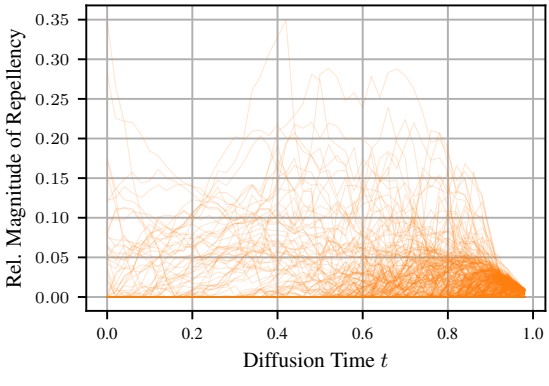

(a) Repellency Strength

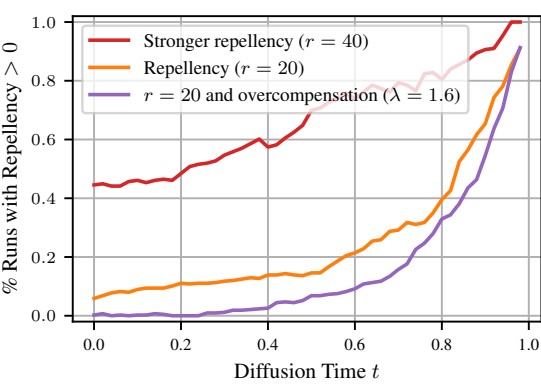

(b) Diffusion steps with repellency

*Figure 5.* (a) The gradient that SPELL adds is only a fraction of the magnitude of the diffusion's score, thus adjusting it without drowning it out. (b) Repellency happens primarily in early backwards steps ($t \in [0.6, 1.0]$) and then remains zero, thus making it sparse. Overcompensation allows finishing the repellency earlier, whereas runs with high repellency radius repel longer. Latent Diffusion trajectories with intra-batch and previous-batch repellency.

that of the diffusion score and never exceeds 35%. This explains why our repellency does not reduce image quality or introduce artifacts. A second reason for this is that SPELL corrections happen mostly in the early stages of generation, which literature claims to be when the rough image is outlined, rather than in late steps, where the image is refined (Biroli et al., 2024; Kynkäänniemi et al., 2024). Recall that the backwards diffusion starts at $t = 1$ and outputs the final image at $t = 0$. Figure 5b shows that at $t = 0.8$, only 40% of the trajectories have a non-zero repellency term, further declining to 21% at $t = 0.6$. If we impose a higher repellency radius $r$, the repellency acts for longer. Especially in this case, adding overcompensation helps. As intended, the repellency strength is increased and in return stops earlier. These stops are often final: The repellency stays zero for the remainder of the generation, verifying that the trajectories do not bounce back into the repellency radii, as shown in Appendix H, and reaffirming SPELL's sparsity.

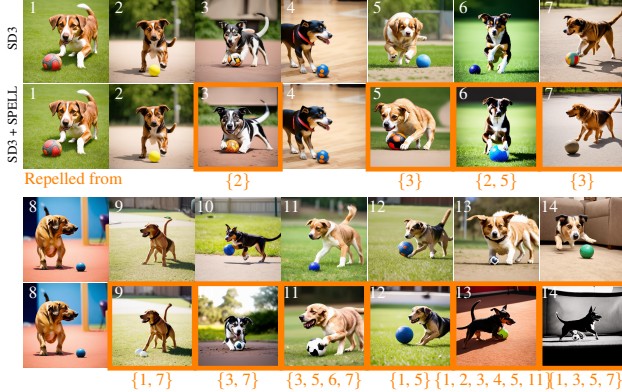

*Figure 6.* High-resolution images for the prompt *"a dog plays with a ball"* generated one-by-one ($B = 1$) with SD3 (top) and SD3 + SPELL (bottom), using the same random seeds. While SPELL does not intervene on the first two images, as they are different enough, it intervenes on the 10 marked with orange borders, as they are too similar to previously generated images. This changes the output from what it would have been without SPELL (at the top) to slightly or completely novel images at the bottom.

## 5.5. Qualitative Example of SPELL Interventions

This section shows what SPELL's correction terms change in practice. Figure 6 shows 14 images in 1024x1024 resolution, generated iteratively using SD3 with and without SPELL interventions. Images are generated one by one ($B = 1$), and when generating the $i + 1$-th image, SPELL repels from the reference set of all images 1 to $i$ it has generated thus-far. Images are highlighted in orange if SPELL enforced changes during their generation trajectory. When SPELL does not intervene, the SD3 + SPELL image (bottom row) coincides with the SD3 output (top row), since we use the same seeds. For images 1 and 2, SPELL did not intervene as it detected that the 2nd trajectory was heading to an image that was novel enough from the 1st. The 3rd image was expected to come out too close to the 2nd at some time during generation, triggering SPELL to alter the background, ball color, and details on the dog. As more images are added to the reference set, SPELL intervenes more often. For example, image 13 is guided away from an image of a dog on a grassy ground with trees in the background, which images 2, 3, 5, and 11 already show, and explores an entirely novel mode, with both a new dog race and a previously unseen surface. A similarly strong intervention happens in image 14. SPELL's sparsity means that it it not always applied, even when there are already many shielded images. This is the case for image 8, which is novel enough to remain unchanged. Note that none of the images has visual artifacts due to SPELL's early and sparse interventions. Appendices J and N present hundreds of images generated with SPELL on further models, affirming this finding.

*Table 2.* Without SPELL, 7.60% of EDMv2's generated images are too close to the protected ImageNet-1k train set. Adding SPELL reduces this rate, and the more inference-runtime is spent on searching Voronoi cells (1, 2, 3, 5, or 10), the better the protection becomes. The runtime is reported on a single A100-40GB GPU.

| Model | Gen. images within shields ↓ | Prec. ↑ | Recall ↑ | Time per image (s) ↓ |
|---|---|---|---|---|
| EDMv2 | 7.60% | 0.792 | 0.242 | 2.43 |
| + SPELL-1 | 1.08% | 0.792 | 0.181 | 4.63 |
| + SPELL-2 | 0.55% | 0.788 | 0.175 | 6.06 |
| + SPELL-3 | 0.33% | 0.777 | 0.162 | 7.79 |
| + SPELL-5 | 0.22% | 0.771 | 0.163 | 9.94 |
| + SPELL-10 | 0.16% | 0.768 | 0.160 | 13.54 |

## 5.6. Image Protection at Scale

Last, we present a second use-case of shielded generation, where the goal is to create images that are sufficiently novel from a given large set of images. In particular, we scale SPELL to shield all 1.2 million ImageNet-1k train images by employing approximate nearest neighbor search (Douze et al., 2024, see Appendix I). We then generate 50k images and track how often they fall into a protected shield.

Table 2 shows that 7.6% of the 50k images that MDTv2 generates without SPELL are within an $L_2$ distance of $r = 60$ of their nearest neighbor on ImageNet. Figure 7 shows examples and verifies that such images are indeed nearly copies of existing images. Adding SPELL with $r = 60$ reduces this rate down to 0.16%. Figure 7 shows that the images are indeed not too close to their ImageNet neighbors anymore. However, the runtime increases. This is not due to SPELL—in all previous experiments with $K = 128$, SPELL does not increase the runtime, see Appendix G—but due to the approximate nearest neighbor search algorithm over the $K = 1.2M$ images that we implement on CPU. Reducing the number of Voronoi cells that the nearest neighbor algorithm searches for shields allows to speed up the generation time, at the cost of a catching less shields. Further improvements in $L_2$ based neighbor search techniques will further increase the protection rate and compute overhead.

We take one final look at precision and recall in this special use-case, where the goal is to generate images that are similar but *not equal* to the train images. Expectedly, the recall over the validation images decreases when repelling from all training images, because validation images may fall into the shield radii around train images. However, the precision remains largely unaffected. This demonstrates again that SPELL stays on the image manifold, even when repelling from many images. Finally, the last two images in Figure 7 give more insight into the workings of the $L_2$ similarity in the VAE latent space that MDTv2 diffuses in. Apparently, image distances inside the VAE space encode a visual similarity where images with similar colors and compositions are close to one another. SPELL could also create shields in se-

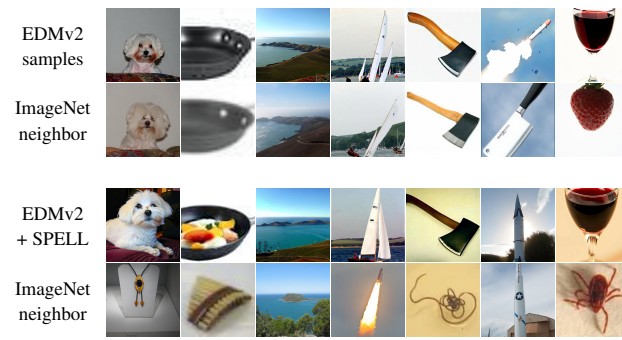

*Figure 7.* The images generated by EDMv2 in the first row are too close to existing images in the ImageNet-1k train set in the second row, which EDMv2 was trained on. SPELL ensures that they maintain a protection radius. The images in third row, generated from the same seeds but with SPELL, are sufficiently different from the ImagetNet neighbors in the second row, and also from their own nearest neighbors in the forth row.

mantic spaces, e.g., by comparing the DINOv2 embeddings of expected image outputs, which we leave for future works.

## 6. Discussion

We introduce sparse repellency (SPELL), a training-free post-hoc method to guide diffusion models *away* from a set of images. SPELL increases diversity by preventing repeat generation, and allows protecting a set of given reference images. SPELL can be applied to any diffusion model, whether it is class-to-image or text-to-image, and whether it is unconditional or classifier(-free) guided, even at high resolution and arbitrarily sized reference sets.

We find three limitations of SPELL: First, theoretically, SPELL currently only guarantees generating images outside the shields if all shields are disjoint. If there are overlapping shields and the diffusion trajectory heads exactly into the middle of them, their repellency terms cancel. While we do not see this problem in practice even at the scale of 1.2M images, in theory, the algorithm could be improved: we can merge overlapping shields, or search for a direction that points out of *the convex hull* of all shields rather than their naive sum of parts, at the cost of scalability. Second, we currently apply SPELL with respect to the $L_2$ distance inside the diffusion models' (VAE encoder) latent spaces, which lead to *visually* different outputs. Checking the distances inside a downstream semantically structured embedding space and propagating their directions back could lead to generating more *semantically* different images. Third, SPELL is applied to expected $\mathbb{E}[\mathbf{X}_0|x_t]$ under the forward joint distribution and not samples from $p_{0|t}$ required for Doob h transform. see Appendix A. This expectation will only correspond to the generative distribution if using the time-reversal diffusion, and not for example when using the probability flow ODE sampler.

## Impact Statement

This paper presents work whose goal is to advance the field of Machine Learning. There are many potential societal consequences of our work, none which we feel must be specifically highlighted here.

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

## A. Guidance via Doob's h-transform

Doob's h-transform provides a definitive approach to conditioning and guiding diffusions. In the context of avoiding points let $S = \cup_k B_k =$ where $B_k = \{x \in \mathcal{X} \ : \ \|x - z_k\|_2 \leq r\}$ are balls of radius $r$ around centers $(z_k)_k$. We may approximate Doob's h transform with some simplifying assumptions based on diffusion posterior sampling (DPS (Chung et al., 2023)). DPS entails approximating $p_{0|t}(x_0 \mid x_t)$ with $\tilde{p}_0(|D(x_t))$, where $D(x_t) = \mathbb{E}[\mathbf{X}_0 \mid x_t]$ for some choice of density $\tilde{p}$.

Let us observe that

$$\nabla \log p_{0|t}(\mathbf{X}_0 \notin \cup_k B_k \mid x_t) = \sum_k \nabla \log p_{0|t}(\mathbf{X}_0 \notin B_k \mid x_t).$$

For simplicity, we approximate the conditional density $p_{0|t}(\cdot \mid x_t)$ with a Gaussian density with mean $\mathbb{E}[\mathbf{X}_0 \mid x_t]$ and variance $\Sigma_t = \mathrm{Id}$. Since, for $X \sim \mathcal{N}(\mu, \mathrm{Id})$, the random variable $\|X - z_k\|^2$ follow a non-centered chi-square distribution $\chi^2_{nc,d}(\lambda)$ where $\lambda := \lambda(\mu) = \|\mu - z_k\|^2$. As such

$$\nabla \log p_{0|t}(\mathbf{X}_0 \notin B_k \mid x_t) \approx \frac{-\nabla_\mu F_{\chi^2_{nc,d}(\lambda)}(r^2)}{1 - F_{\chi^2_{nc,d}(\lambda)}(r^2)} = (\mu - z_k) \times \omega(\lambda(\mu), r)$$

with weights function

$$\omega(\lambda, r) = \frac{2}{F_{\chi^2_{nc,d}(\lambda)}(r^2) - 1} \times \frac{\partial F_{\chi^2_{nc,d}(\lambda)}}{\partial \lambda}(r^2).$$

We recall that the CDF of $\chi^2_{nc,d}$ is a combination of the CDF of standard $\chi^2_d$:

$$F_{\chi^2_{nc,d}(\lambda)}(x) = \sum_{j=0}^\infty c_j(\lambda) F_{\chi^2_{d+2j}}(x)$$

$$\frac{\partial F_{\chi^2_{nc,d}(\lambda)}}{\partial \lambda}(x) = \frac{1}{2} \sum_{j=0}^\infty c_j(\lambda) \left[ F_{\chi^2_{d+2(j+1)}}(x) - F_{\chi^2_{d+2j}}(x) \right]$$

$$\nabla_\mu F_{\chi^2_{nc,d}(\lambda)}(x) = \frac{\partial F}{\partial \lambda}(x) \times \partial_\mu \lambda(\mu) = \frac{\partial F}{\partial \lambda}(x) \times 2(\mu - z_k)$$

where we denoted $c_j(\lambda) = \frac{(\lambda/2)^j e^{-\lambda/2}}{j!}$.

## B. Repellence Guarantee

Consider again our adjusted SDE

$$dx = \left[ f(x,t) - g^2(t) \left( \nabla_x \log p_t(x) + \frac{\alpha_t}{\sigma_t^2} \sum_{k=1}^{K} (\hat{x}_0 - z_k) \text{ReLU}\left( \frac{r}{||\hat{x}_0 - z_k||_2} - 1 \right) \right) \right] dt + g(t)dw,$$

where $\hat{x}_0 := \frac{x_t + \sigma_t^2 \nabla_x \log p_t(x)}{\alpha_t}$ is the expected image if we did not intervene.

This section shows that the SDE leads to a output distribution with $P_0(\{B_r(z_k)|k=1,\ldots,K\}) = 0$. This ensures that it does not create samples within radius $r$ around the repellence images $z_k, k=1,\ldots,K$. To this end, assume we have a set of repellence images and that their repellence balls do not overlap (otherwise, one can merge them and select an according higher radius). Let's consider an arbitrary timestep $t$. Then Tweedie's formula (Efron, 2011; Bradley & Nakkiran, 2024) gives that

$$
\begin{aligned}
\mathbb{E}[\mathbf{X}_0|x_t] &= \frac{x_t + \sigma_t^2 \left( \nabla_x \log p_t(x) + \frac{\alpha_t}{\sigma_t^2} \sum_{k=1}^{K} (\hat{x}_0 - z_k) \text{ReLU}\left( \frac{r}{||\hat{x}_0-z_k||_2} - 1 \right) \right)}{\alpha_t} \\
&= \frac{x_t + \sigma_t^2 \nabla_x \log p_t(x)}{\alpha_t} + \frac{\sigma_t^2}{\alpha_t} \frac{\alpha_t}{\sigma_t^2} \sum_{k=1}^{K} (\hat{x}_0 - z_k) \text{ReLU}\left( \frac{r}{||\hat{x}_0-z_k||_2} - 1 \right) \\
&= \hat{x}_0 + \sum_{k=1}^{K} (\hat{x}_0 - z_k) \text{ReLU}\left( \frac{r}{||\hat{x}_0 - z_k||_2} - 1 \right)
\end{aligned}
$$

**Case 1:** $||\hat{x}_0 - z_k||_2 \ge r \,\forall k = 1,\ldots,K$. Then the ReLU term becomes 0 and $\hat{x}_0$ remains unadjusted and $||\mathbb{E}[\mathbf{X}_0|x_t] - z_k||_2 \ge r$.

**Case 2:** $\exists k^\star \in \{1,\ldots,K\} : ||\hat{x}_0 - z_k^*||_2 < r$. Since the balls are non-overlapping,

$$\sum_{k=1}^{K} (\hat{x}_0 - z_k) \text{ReLU}\left( \frac{r}{||\hat{x}_0 - z_k||_2} - 1 \right) = (\hat{x}_0 - z_k^*) \text{ReLU}\left( \frac{r}{||\hat{x}_0 - z_k^*||_2} - 1 \right).$$

Then

$$
\begin{aligned}
||\mathbb{E}[\mathbf{X}_0|x_t] - z_k||_2 &= ||\hat{x}_0 + (\hat{x}_0 - z_k^*) \text{ReLU}\left( \frac{r}{||\hat{x}_0 - z_k^*||_2} - 1 \right) - z_k^*||_2 \\
&= ||(\hat{x}_0 - z_k^*) + (\hat{x}_0 - z_k^*) \left( \frac{r}{||\hat{x}_0 - z_k^*||_2} - 1 \right)||_2 \\
&= ||(\hat{x}_0 - z_k^*) + (\hat{x}_0 - z_k^*) \frac{r}{||\hat{x}_0 - z_k^*||_2} - (\hat{x}_0 - z_k^*)||_2 \\
&= ||r \frac{(\hat{x}_0 - z_k^*)}{||\hat{x}_0 - z_k^*||_2}||_2 \\
&= r
\end{aligned}
$$

So, in all cases, $||\mathbb{E}[\mathbf{X}_0|x_t] - z_k||_2 \ge r$, for any $t$. Especially, for $t = 0$, the SDE does not add any noise anymore and the sampled $x_0$ is equal to the expectation.

Hence $||x_0 - z_k||_2 \ge r \,\forall k = 1,\ldots,K$.

## C. Conservative Field Interpretation

The function $h(x) = \sigma_{\mathrm{relu}}\left(\frac{r}{\|x\|} - 1\right)x$ is the conservative field associated to the (family of) potential $H : \mathbb{R}^d \to \mathbb{R}$:

$$H(x) = \begin{cases} r\|x\| - \frac{1}{2}\|x\|^2 & \text{when } \|x\| < r, \\ \frac{r^2}{2} & \text{otherwise,} \end{cases} \tag{8}$$

where Gauge $H(0)$ is chosen arbitrarily, as illustrated in Figure 8.

Furthermore, observe that the mapping $x_t \mapsto \hat{x}_0$ defined by $\hat{x}_0 = \frac{1}{\alpha_t}\left(\sigma_t^2 \nabla \log p_t(x_t) + x_t\right)$ is a conservative field given by the potential $\frac{1}{\alpha_t}\left(\sigma_t^2 \log p_t(x_t) + \frac{1}{2}\|x_t\|^2\right)$.

Therefore, SPELL guidance in Equation 5 is the composition of two conservative fields. But note that, in general, conservative fields are not stable by composition, unless the Hessians of their potentials commute everywhere.

**Theorem C.1.** *We consider $f : \mathbb{R}^d \to \mathbb{R}$ a twice differentiable function. The Jacobian of the map $\phi : x \mapsto \frac{\nabla f(x)}{\|\nabla f(x)\|}$ is given by*

$$\mathrm{Jac}(\phi)(x) = \frac{1}{\|g\|}H - \frac{1}{\|g\|^3}gg^T H, \text{ with } g = \nabla f(x) \text{ and } H = \nabla^2 f(x)$$

Hence, in the case where $H$ and $gg^T$ commute, this Jacobian is locally symmetric. If they commute everywhere, then this Jacobian is globally symmetric, and $\phi$ is a conservative field.

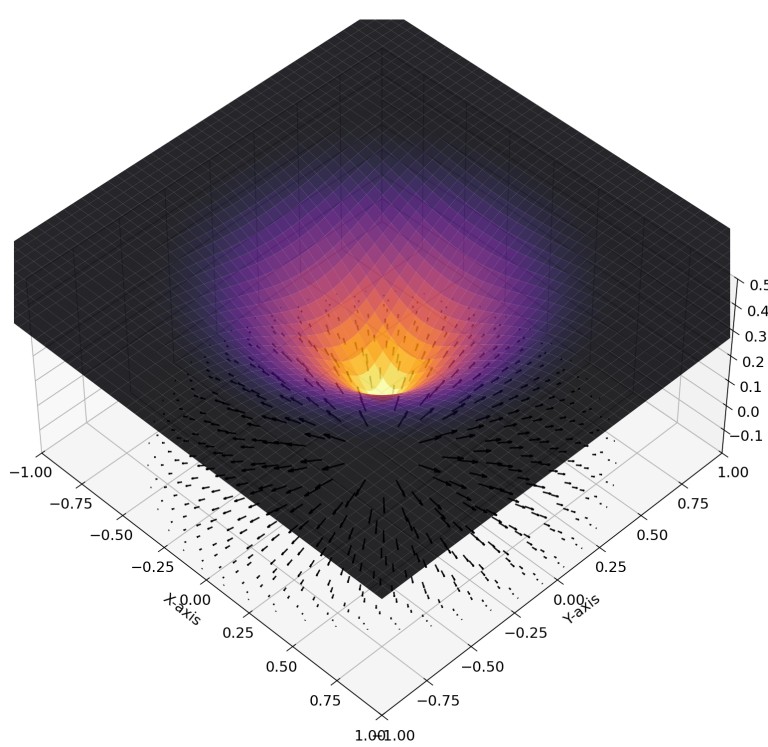

*Figure 8.* Potential function whose gradient field is $\sigma_{\mathrm{relu}}\left(\frac{r}{\|x\|} - 1\right)x$, displayed for $x \in \mathbb{R}^2$. Repellence force is dynamic: closer to the center (i.e., when a diffusion trajectory is expected to be close to a protected image) it applies stronger gradients, as shown by the arrows, while outside the repellecy radius, it applies no gradient at all, letting the diffusion trajectory continue without any intervention.

## D. Implementation Details and Hyperparameters

Since SPELL is a training-free post-hoc method, we use the trained checkpoints of diffusion models provided by their original authors. For EDMv2 and MDTv2, we use the hyperparameters suggested by their authors. Latent Diffusion, Simple Diffusion, and Stable Diffusion come without recommended hyperparameters, so we tune the classifier-free guidance (CFG) weight by the F-score between precision and coverage on the 554 validation captions on our CC12M split.

For the repellence radius $r$, the latent spaces that the different models diffuse on have different dimensionalities, hence the scales of the repellence radii differ. To get a sense of the scales, we first generate one batch of images without repellence and tracked the pairwise $L_2$ distance between generated latents at the final timestep. We then test 16 values from 0 to 2 times the median distance. This yields the following hyperparameters for the results in Table 1.

**EDMv2:** CFG weight 1.2, 50 backwards steps, $\sigma_{\min} = 0.002$, $\sigma_{\max} = 80$, $\rho = 7$, $S_{\min} = 0$, $S_{\max} = \infty$, repellence radius $r = 20$, batchsize 8.

**MDTv2:** CFG weight 3.8, 50 backwards steps, repellence radius $r = 45$, batchsize 2.

**Stable Diffusion 3:** CFG weight 5.5, 28 backwards steps, repellence radius $r = 200$, on CC12M overcompensation 1.6 (no overcompensation on ImageNet), batchsize 8.

**Simple Diffusion:** CFG weight 5.5, 50 backwards steps, repellence radius $r = 50$, overcompensation 1.6, batchsize 16.

**Latent Diffusion:** CFG weight 5, 50 backwards steps, repellence radius $r = 20$, overcompensation 1.6, batchsize 8.

Algorithm 1 gives a high-level pseudo-code for SPELL and Algorithm 2 details how we implemented SPELL in a parallelized way in Python.

---

**Algorithm 1** SPELL added to the backwards diffusion step. This is a simplified example, see Appendix D for Python code that is parallelized and supports sparse neighbor retrieval.

---

**Input:** *Batch of latents* $\{x_t^{(b)}\}_{b=1,\ldots,B}$, *set of shielded images* $\{z_k\}_{k=1,\ldots,K}$, *radius* $r$, $\lambda$  **for** $b = 1, \ldots, B$ **do**

$\quad\hat{x}_0^{(b)} \leftarrow D_{\theta^\star}(t, x_t^{(b)})$       ▷ *Expected diffusion output without repellency*

**end**

**for** $b = 1, \ldots, B$ **do**

$\quad\vec{\Delta}_b \leftarrow 0$  **for** $k = 1, \ldots, K$ **do**       ▷ *Repel from the shielded set*

1    **if** $\|\hat{x}_0^{(b)} - z_k\|_2 < r$ **then**

2     $\vec{\Delta}_b = \vec{\Delta}_b + (\hat{x}_0^{(b)} - z_k)\,\mathrm{ReLU}\left(\frac{r}{\|\hat{x}_0^{(b)} - z_k\|_2} - 1\right)$

3    **end**

4   **end**

5   **for** $b' = 1, \ldots, B, b' \neq b$ **do**       ▷ *Repel within the batch*

6    **if** $\|\hat{x}_0^{(b)} - \hat{x}_0^{(b')}\|_2 < r$ **then**

7     $\vec{\Delta}_b = \vec{\Delta}_b + (\hat{x}_0^{(b)} - \hat{x}_0^{(b')})\,\mathrm{ReLU}\left(\frac{r}{\|\hat{x}_0^{(b)} - \hat{x}_0^{(b')}\|_2} - 1\right)$

8    **end**

9   **end**

10   Calculate $x_{t-1}^{(b)}$ by taking a step towards $\hat{x}_0^{(b)} + \lambda\vec{\Delta}_b$ (using the diffusion scheduler)

**end**

**Output:** $\{x_{t-1}^{(b)}\}_{b=1,\ldots,B}$

---

## E. Construction of the Soft-label CC12M Dataset

CC12M is a recent text-to-image dataset that contains pairs of image links and the title scraped from their metadata. To turn this into our soft-label subset of CC12M, where each caption has a set of multiple possible images related to it, we first group all images in CC12M by their caption and keep only captions with at least four and at most 128 images.

Some of these images are falsely grouped together. For example, there are photo albums whose images were assigned the

```python
def backward_step(x_t, t, protection_set, r, lambda, repel_within_batch):
    """
    A generation step from t to t-1 of a diffusion with repellency.

    x_t: Matrix of size [batch, dimensions] containing the current latents
    t: float, current time
    protection_set: Either a matrix of size [num_protection_images, dimensions] with latents we want to repel from
     or a database that will output closest neighbors in this format
    radius: Float, repellency radius
    lambda: Float, overcompensation factor
    repel_within_batch: Boolean, whether to apply intra-batch repellency
    """

    # Predict x_0 using the diffusion model (using diffusion without repellency)
    x_0_hat = diffusion_score.predict(x_t, t)

    # Repel from protection set
    repellency_term = 0
    if protection_set is not None:
        if isinstance(protection_set, database):
            protection_set = protection_set.find_neighbors_within_radius(x_0_hat, radius)
        diff_vec = x_0_hat.unsqueeze(1) - mu.unsqueeze(0)
        # diff_vec has size [batch, num_protection_images, dimensions]
        weight = (diff_vec**2).sum(dim=2).sqrt()
        trunc_weight = ReLU(radius / diff - 1)
        repellency_term += (diff_vec * trunc_weight).sum(dim=1)

    # Repel within batch
    if repel_within_batch:
        diff_vec = x_0_hat.unsqueeze(1) - x_0_hat.unsqueeze(0)
        # diff_vec has size [batch, batch, dimensions]
        weight = (diff_vec**2).sum(dim=2).sqrt()
        trunc_weight = ReLU(radius / diff - 1)
        diag(trunc_weight) = 0   # Don't repel from the image itself
        repellency_term += (diff_vec * trunc_weight).sum(dim=1)

    # Add our repellency term to the current x_0_hat prediction
    x_0_hat = x_0_hat + lambda * repellency_term

    # Step from t to t-1 using the diffusion update rule (same as in typical diffusion)
    x_t_minus_1 = calculate_update(x_0_hat, x_t, t)
    if t > 0:
        x_t_minus_1 += generate_noise(t)

    return x_t_minus_1
```

Algorithm 2: Our repellency can be added to the backwards algorithm of existing diffusion models, without retraining. Since the expected x_0_hat is often already computed as part of the backward process, the only runtime overhead are the pairwise differences and the possible neighbor search.

same generic title in their metadata. A useful heuristic to filter out such cases is to analyze the top-level domains of the images. We filter out sets where the most frequent top-level domain belongs to $75\%$ or more of the image urls. Second, we filter out automatically generated captions by removing captions that include the strings 'Display larger image', 'This image may contain', 'This is the product title', or 'Image result for'. Last, due to privacy guidelines, we filter out any caption whose image may include individuals. This is done by filtering out caption that include '<PERSON>', which is a placeholder that the CC12M dataset overwrote any possible person name with. After these filtering steps, we arrive at 5554 captions. We randomly split them into a validation set of 554 captions and a test set of 5000 captions. Table 3 shows how many images belong to each caption.

We did not filter any images out although there are some near-duplicates. This is done on purpose in order to not skew the distributions. Filtering out captions amounts to deciding on which subset of the dataset we test our models on. But filtering out images would change the conditional distributions $P(X|c)$ to something different from the training distributions. In other words, a model that learned the train distribution ideally is expected to have a stronger mode at near-duplicate images but testing it on a changed $P(X|c)$ distribution would punish it for learning the correct distribution. If a future work intends to test models on unseen images, we note that removing near-duplicates may be a possibility, depending on the experiment design.

*Table 3.* Number of captions that have a certain number of images attached to them in our soft-label CC12M dataset.

| Images per caption | Validation split | Test split |
|---|---|---|
| 4 − 5 | 270 | 2600 |
| 6 − 10 | 174 | 1485 |
| 11 − 20 | 75 | 555 |
| 21 − 30 | 20 | 219 |
| 31 − 40 | 10 | 86 |
| 41 − 50 | 3 | 32 |
| 51 − 128 | 2 | 23 |

## F. Further Diversity-quality Tradeoffs

In addition to the tradeoff experiments in Section 5.3, Table 4 provides the full combinations of metrics attainable with each method, depending on how one chooses the hyperparameters. This is the raw data underlying Figure 4 and allows the curious reader to compare arbitary tradeoffs.

## G. Runtime Analysis and Comparison

The scale of the overhead that SPELL adds is negligible when contrasted with the diffusion generation cost. It amounts to computing (up to) $[B, K]$, distance matrices per time $t$, where both the batchsize $B$ and the size of the protection set $K$ do not exceed hundreds, and adding one single correction vector to the score. Table 5 confirms that the runtime that SPELL adds (as well as the other benchmarked diversity methods) is negligible, here using $B = 8$ and intra-batch repellency, hence $K = B - 1 = 7$. This also further confirms that the runtime observed in Section 5.6 is due to the next-neighbor search algorithm, not SPELL's correction terms.

## H. Ablation: Repellency Strength Throughout the Generation

In this section, we scrutinize how and when repellency acts during the generation. We also use these insights to run ablations that foster the intuition on the role of the repellence radius.

To begin with, Figure 9 shows repellency in the standard setting with a repellency radius of 25 in Latent Diffusion. We first generate 8 images per prompt, and then generate another 8 images that repel from the first ones, without intra-batch repellency. Figure 9a shows how high the $L_2$ norm of the total gradient is that our repellency adds to the score, divided by the $L_2$ norm of the score. It can be seen that the repellency term is in most cases at most 20% as strong as the original diffusion gradient field. Intuitively, this means that our repellence does not drown out the diffusion model, but is more a corrective term. Repellency mostly takes place early in the backwards diffusion ($t \in [0.6, 1.0]$), with Figure 9b demonstrating that more than 50% of the generations have already finished their repellency in the first quarter of timesteps (note that Latent Diffusion uses linearly scheduled timesteps). This leaves sufficient time for the diffusion model to generate high quality images in the remainder of steps.

Figure 10 uses intra-batch repellency instead of repelling from 8 previously generated images. The dynamics are very similar to Figure 9 (see also the comparison in Figure 15). This shows that our repellency smoothly can be used both intra-batch or iteratively, or in a mixture of both, to generate arbitrary amounts of diverse data even when GPU memory is limited. This mixed setup is presented in Figure 11, where we generate two images at a time that repel both intra-batch and from the previous images. It behaves similarly in both magnitude and duration of repellency. Figure 12 further investigates scalability. Despite repelling from 64 previously generated images, the repellency magnitudes and times are only slightly increased compared to Figure 9. Note that this is despite generating 64+8 images conditionally on the same prompt, repellency from a dataset of more various images like in Section 5.6 is even less effected.

If repellency needs to protect a large radius, the repellency takes place longer in the backwards diffusion process, as shown in Figure 13, where we use an increased radius of 37.5. Here, 43% of the backwards diffusions apply repellency until the end of the generation. The repellency magnitude is increased but still stays below 50% of the magnitude of the diffusion score. One option to speed up the repellency if it runs until the end like here is overcompensation. Figure 14 shows that compared to Figure 9, the repellency is stronger at start and manages to push the trajectories into the diffusion cones of different modes, in return allowing to stop the repellency earlier. This implies that overcompensation can also be used as a

*Table 4.* Metrics of all approaches in the tradeoff experiments in Figure 4.

| Method | Recall | Vendi Score | Coverage | Precision | Density | FID | $FD_{\text{DINOv2}}$ | CLIP Score |
|---|---|---|---|---|---|---|---|---|
| Base Model | 0.237 | 2.527 | 0.446 | 0.558 | 0.768 | 9.566 | 105.967 | 27.789 |
| Particle Guidance, strength = 1024 | 0.099 | 1.987 | 0.249 | 0.300 | 0.326 | 84.115 | 705.661 | 24.470 |
| Particle Guidance, strength = 512 | 0.230 | 2.753 | 0.378 | 0.443 | 0.534 | 23.106 | 286.093 | 26.740 |
| Particle Guidance, strength = 256 | 0.252 | 2.656 | 0.429 | 0.523 | 0.682 | 11.934 | 154.897 | 27.440 |
| Particle Guidance, strength = 128 | 0.248 | 2.591 | 0.447 | 0.553 | 0.754 | 9.442 | 109.257 | 27.704 |
| Particle Guidance, strength = 64 | 0.245 | 2.561 | 0.449 | 0.559 | 0.771 | 9.072 | 101.796 | 27.781 |
| Particle Guidance, strength = 32 | 0.235 | 2.528 | 0.445 | 0.557 | 0.763 | 9.724 | 108.382 | 27.812 |
| Particle Guidance, strength = 16 | 0.236 | 2.529 | 0.446 | 0.557 | 0.764 | 9.596 | 107.041 | 27.813 |
| Interval Guidance, [0.1,0.9] | 0.372 | 2.840 | 0.455 | 0.537 | 0.730 | 8.385 | 85.871 | 27.453 |
| Interval Guidance, [0.2,0.9] | 0.419 | 2.994 | 0.442 | 0.514 | 0.689 | 8.359 | 85.094 | 26.813 |
| Interval Guidance, [0.1,0.8] | 0.470 | 3.174 | 0.448 | 0.500 | 0.663 | 7.507 | 76.104 | 27.215 |
| Interval Guidance, [0.3,0.9] | 0.471 | 3.208 | 0.421 | 0.483 | 0.635 | 8.406 | 87.971 | 25.885 |
| Interval Guidance, [0.2,0.8] | 0.518 | 3.340 | 0.434 | 0.478 | 0.624 | 7.478 | 75.250 | 26.544 |
| Interval Guidance, [0.1,0.7] | 0.567 | 3.576 | 0.432 | 0.451 | 0.577 | 6.804 | 72.092 | 26.784 |
| Interval Guidance, [0.4,0.9] | 0.525 | 3.495 | 0.395 | 0.442 | 0.569 | 8.623 | 96.611 | 24.630 |
| Interval Guidance, [0.3,0.8] | 0.571 | 3.575 | 0.411 | 0.446 | 0.570 | 7.556 | 78.887 | 25.549 |
| Interval Guidance, [0.2,0.7] | 0.614 | 3.770 | 0.417 | 0.426 | 0.536 | 6.771 | 72.972 | 25.979 |
| Interval Guidance, [0.1,0.6] | 0.673 | 4.138 | 0.396 | 0.385 | 0.466 | 6.885 | 81.643 | 26.020 |
| CADS, mixture factor = 0, $\tau_1 = 0.6$ | 0.262 | 2.598 | 0.447 | 0.553 | 0.753 | 9.248 | 105.006 | 27.746 |
| CADS, mixture factor = 0, $\tau_1 = 0.7$ | 0.253 | 2.579 | 0.448 | 0.555 | 0.757 | 9.288 | 105.549 | 27.757 |
| CADS, mixture factor = 0, $\tau_1 = 0.8$ | 0.245 | 2.561 | 0.449 | 0.557 | 0.762 | 9.356 | 105.856 | 27.771 |
| CADS, mixture factor = 0, $\tau_1 = 0.9$ | 0.239 | 2.545 | 0.450 | 0.559 | 0.767 | 9.452 | 106.455 | 27.790 |
| CADS, mixture factor = 0.001, $\tau_1 = 0.6$ | 0.325 | 2.816 | 0.442 | 0.531 | 0.696 | 8.897 | 105.081 | 27.534 |
| CADS, mixture factor = 0.001, $\tau_1 = 0.7$ | 0.297 | 2.734 | 0.446 | 0.540 | 0.719 | 8.963 | 104.006 | 27.617 |
| CADS, mixture factor = 0.001, $\tau_1 = 0.8$ | 0.277 | 2.660 | 0.447 | 0.548 | 0.739 | 9.098 | 103.766 | 27.697 |
| CADS, mixture factor = 0.001, $\tau_1 = 0.9$ | 0.256 | 2.588 | 0.448 | 0.554 | 0.755 | 9.273 | 105.268 | 27.754 |
| CADS, mixture factor = 0.002, $\tau_1 = 0.6$ | 0.425 | 3.208 | 0.417 | 0.472 | 0.584 | 9.870 | 129.159 | 26.920 |
| CADS, mixture factor = 0.002, $\tau_1 = 0.7$ | 0.380 | 3.028 | 0.429 | 0.501 | 0.637 | 9.143 | 114.333 | 27.242 |
| CADS, mixture factor = 0.002, $\tau_1 = 0.8$ | 0.330 | 2.837 | 0.442 | 0.529 | 0.692 | 8.893 | 105.511 | 27.506 |
| CADS, mixture factor = 0.002, $\tau_1 = 0.9$ | 0.277 | 2.660 | 0.446 | 0.548 | 0.739 | 9.098 | 103.762 | 27.696 |
| SPELL, shield radius = 40 | 0.370 | 2.998 | 0.437 | 0.500 | 0.631 | 13.072 | 140.841 | 27.397 |
| SPELL, shield radius = 35 | 0.359 | 2.935 | 0.445 | 0.518 | 0.665 | 11.452 | 120.346 | 27.556 |
| SPELL, shield radius = 30 | 0.337 | 2.856 | 0.451 | 0.531 | 0.695 | 10.349 | 106.753 | 27.655 |
| SPELL, shield radius = 25 | 0.312 | 2.774 | 0.454 | 0.542 | 0.723 | 9.794 | 100.123 | 27.739 |
| SPELL, shield radius = 20 | 0.287 | 2.691 | 0.455 | 0.552 | 0.746 | 9.535 | 98.666 | 27.781 |
| SPELL, shield radius = 15 | 0.263 | 2.616 | 0.454 | 0.558 | 0.762 | 9.558 | 100.709 | 27.811 |

means to realize higher repellency radii, without needing to repel until $t = 0$. We leave this, and possibly expansions with overcompensation or repellency radius schedulers, for future works.

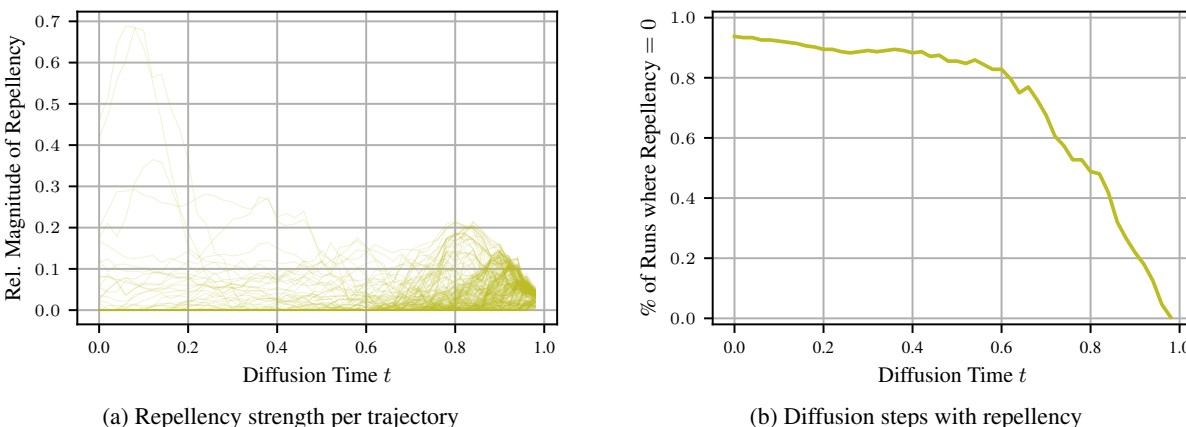

(a) Repellency strength per trajectory          (b) Diffusion steps with repellency

*Figure 9.* Generating images that repel from 8 protected images (generated with the same prompt). Latent Diffusion, 256 generations in total.

*Table 5.* Generation times per image. Neither SPELL nor other diversity inducing methods add considerable runtime. The runtime is dominated by the diffusion backbone. Mean $\pm$ standard deviation across 500 images, run on an NVIDIA V100 GPU.

| Model | Generation time per image (seconds) |
|---|---|
| Baseline (Simple Diffusion) | $2.93 \pm 0.12$ |
| Simple Diffusion + PG | $2.96 \pm 0.13$ |
| Simple Diffusion + IG | $2.93 \pm 0.12$ |
| Simple Diffusion + CADS | $2.96 \pm 0.12$ |
| Simple Diffusion + SPELL | $2.94 \pm 0.13$ |

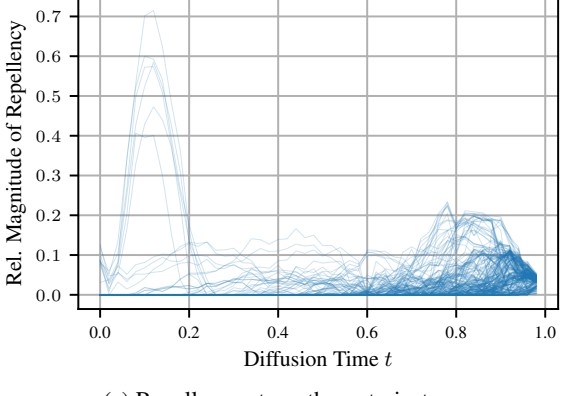 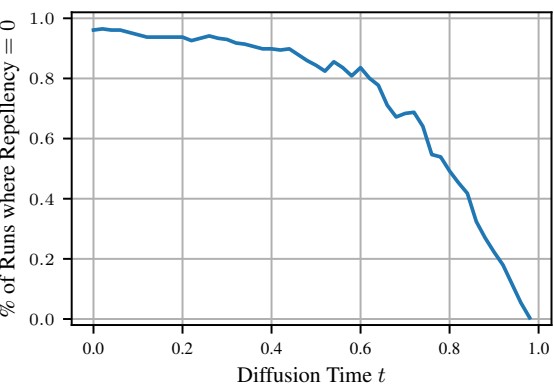

(a) Repellency strength per trajectory

(b) Diffusion steps with repellency

*Figure 10.* Generating images with the same prompt in batches of 8 with intra-batch repellency. Latent Diffusion, 256 generations in total.

# I. Image Protection on Large Datasets

Image protection involves computing the repellence between the current batch $x_t$ being generated with a large dataset $\mathcal{D}$ of size $N$, with $N \gg 10^5$. This dataset will be typically too large to fit entirely in GPU memory. Furthermore, computing the repellence term of each element of the batch with every element of the dataset would be prohibitive. However, since the repellence term is zero for vectors that are far-away, this opens the possibility of an optimization: first, the *closest* images from the batch are retrieved using a vector similarity index (stored in RAM), and only then these images are moved into GPU memory for the actual computation of the repellence term. An efficient implementation of this technique is provided by the Faiss library (Douze et al., 2024). We use the IndexIVFFlat object, that rely on Voronoi cells to cluster vectors and speed-up search. We chose a number of Voronoi cells equal to the square root of dataset size, i.e 1131 cells containing typically 1132 examples each. During generation, we probe only the two voronoi cells closest to the current expected outputs. The behavior of the repellence term ensures that false positive are rarely a problem. False negatives (if any) are typically "far-away" which means that their contribution to the sum of all ReLU repellency terms would have been small. In Table 2, we show that one Voronoi cell is often enough. Searching the ten closest cells gives an even higher protection rate, though at the cost of higher searching costs. This shows that advances in efficient search algorithms will directly benefit SPELL when it is applied to large repellency sets.

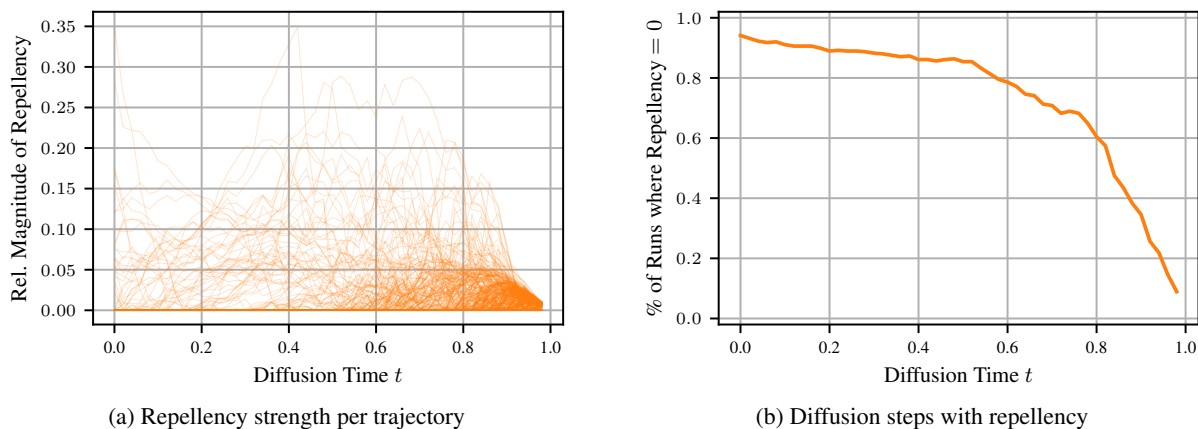

(a) Repellency strength per trajectory

(b) Diffusion steps with repellency

*Figure 11.* Generating images by iteratively, generating 2 images at a time. They repel both intra-batch and from the previously generated images. We use 50 different prompts, generating 4-32 images each, giving a realistic setup. Latent Diffusion, 452 generations in total.

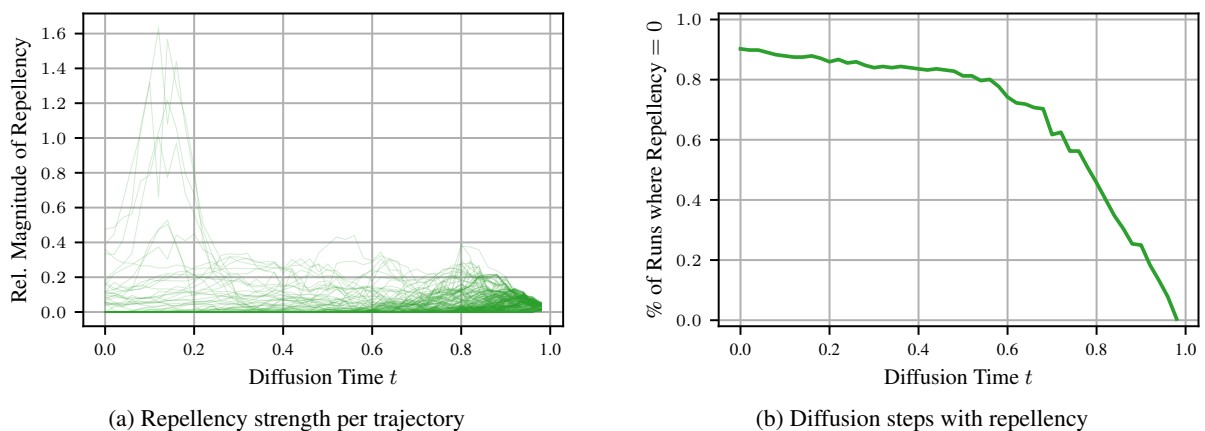

(a) Repellency strength per trajectory

(b) Diffusion steps with repellency

*Figure 12.* Generating images that repel from 64 protected images (generated with the same prompt). Latent Diffusion, 256 generations in total.

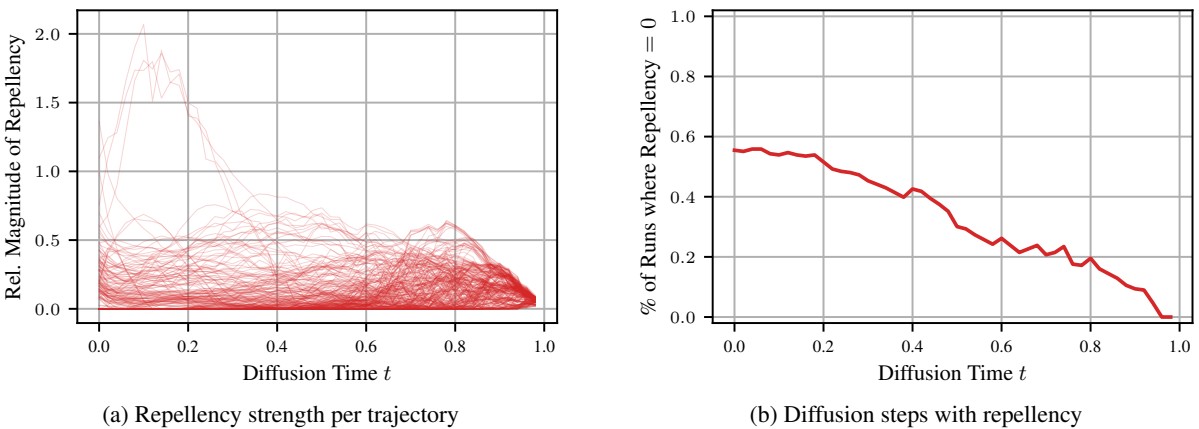

(a) Repellency strength per trajectory

(b) Diffusion steps with repellency

*Figure 13.* Generating images that repel from 8 protected images (generated with the same prompt), using a 1.5 times larger repellency radius. Latent Diffusion, 256 generations in total.

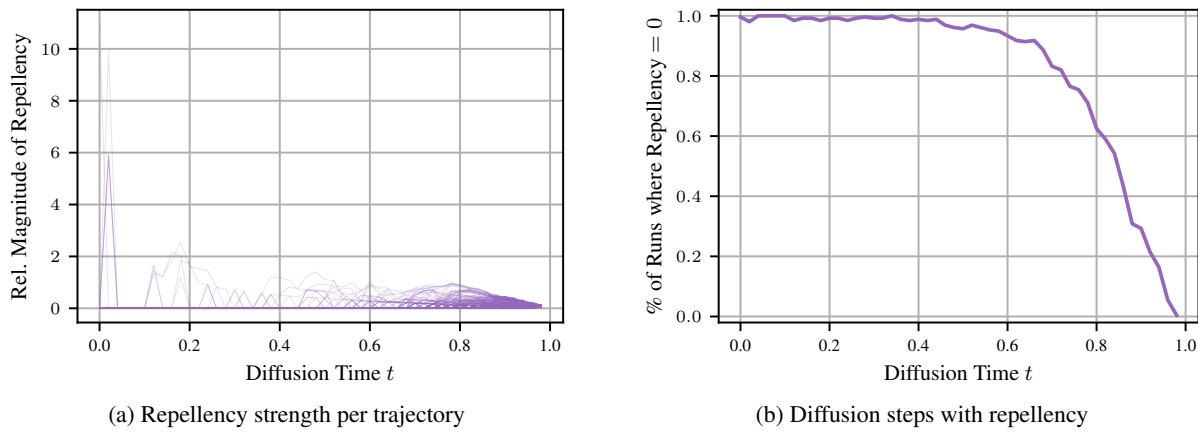

(a) Repellency strength per trajectory

(b) Diffusion steps with repellency

*Figure 14.* Generating images that repel from 8 protected images (generated with the same prompt), with an overcompensation factor of 2. Latent Diffusion, 256 generations in total.

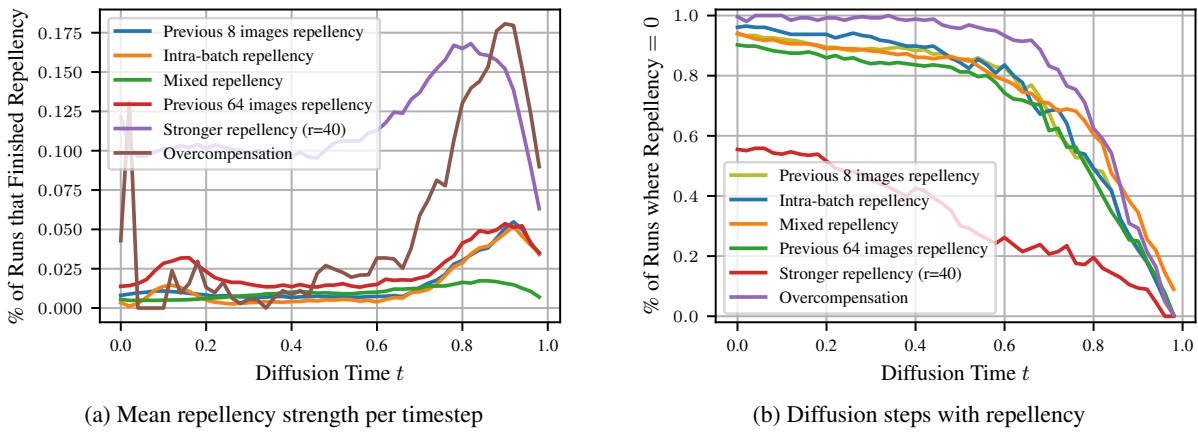

(a) Mean repellency strength per timestep

(b) Diffusion steps with repellency

*Figure 15.* Comparison of the previous ablations.

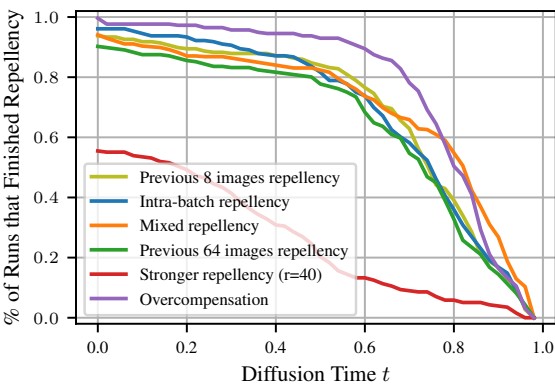

*Figure 16.* Timesteps at which the repellency has finished, in that the term is zero and stays zero for the remainder of the generation.

## J. Examples of Images Generated with Repellency

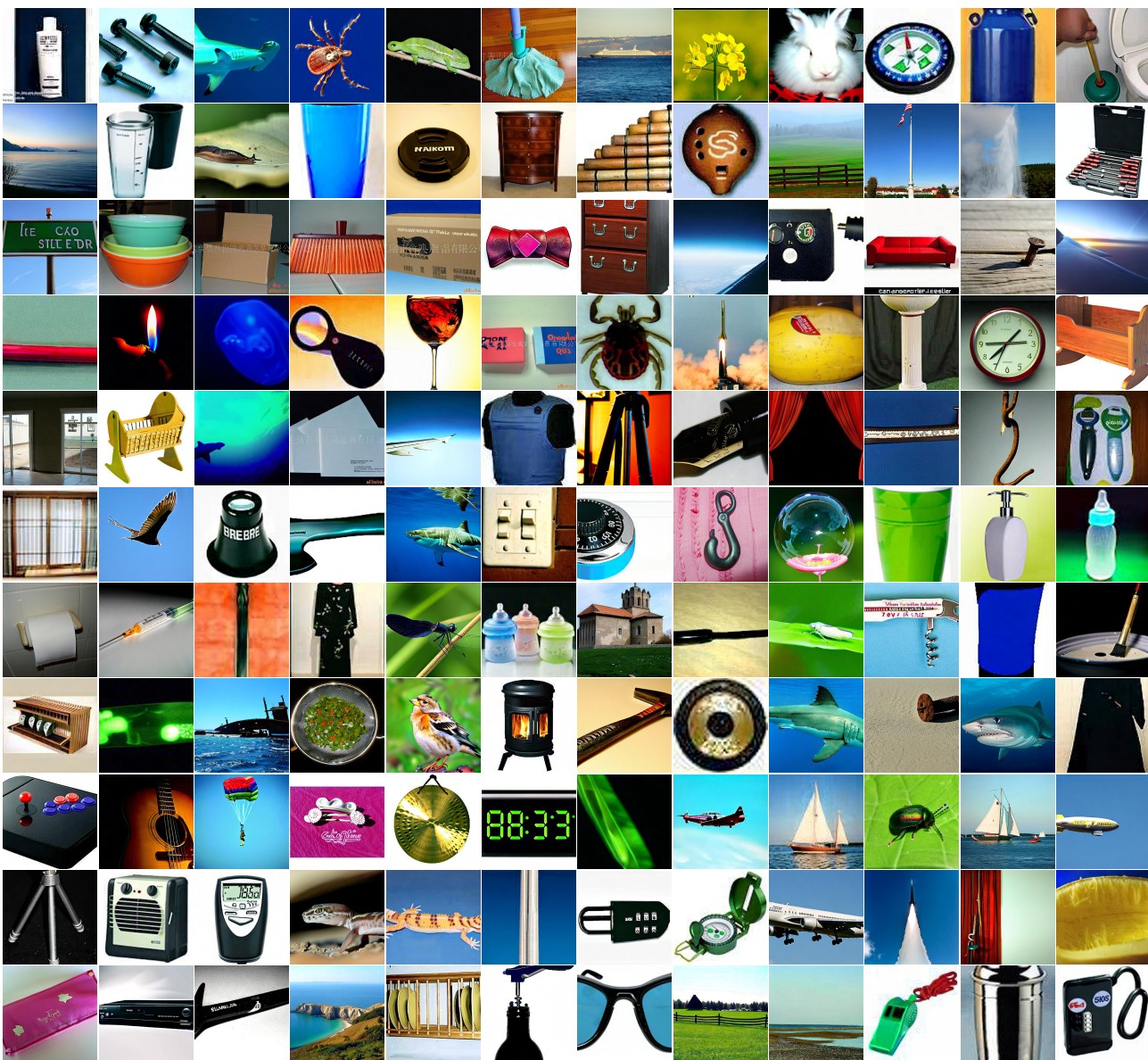

*Figure 17.* Randomly chosen images where repellency actively pushed EDMv2 away from the protected ImageNet-1k train set in Section 5.6. All images have repellency applied to them but do not show visual artifacts. Low-quality images are by design because the underlying EDMv2 model learned to generate this style of images from the ImageNet-1k train dataset.

## K. Ablation: SPELL on Different Prompt Lengths

To verify that SPELL does not only increase the diversity of short prompts, where it is easier to find different images fitting to a prompt, we stratify our analysis by prompt length. Fig. 18 shows that the SPELL increases the diversity, here of the Latent Diffusion model, throughout all prompt length quantiles. Notably, this is on a relatively small repellency radius $r$, which does not decrease the prompt-adherence as measured by the CLIP score.

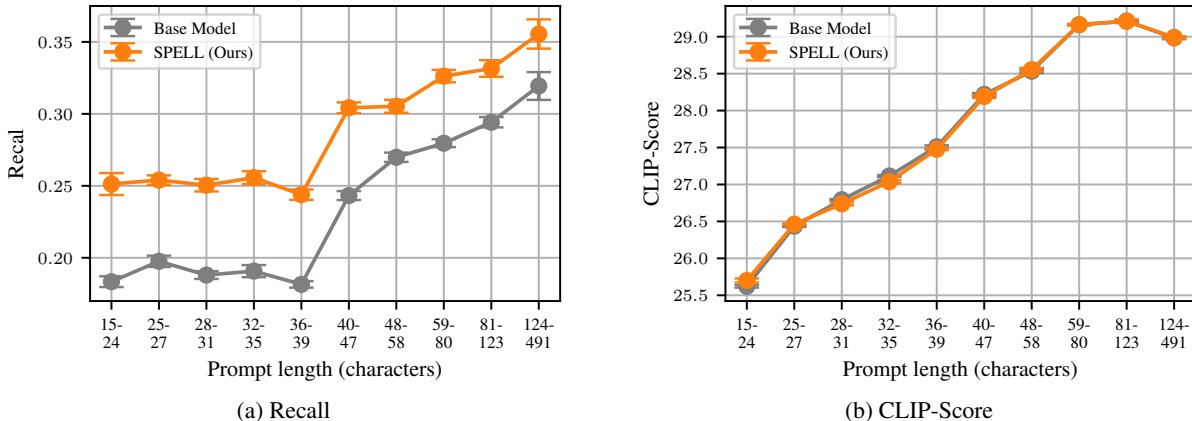

(a) Recall

(b) CLIP-Score

*Figure 18.* Diversity and prompt-adherence for short and long prompts. We split the CC12M prompts into 10 categories based on the number of characters. SPELL achieves a consistently higher diversity than the baseline Latent Diffusion model both for short and long prompts, while maintaining the same CLIP-Score as the baseline model. An example for a short prompt is "Bird on a tree branch" whereas long prompts include "Head Medusa, creature of Greek mythology. pieces made by hand with goldsmiths and metals such as gold and copper. wears a helmet of green and gold snakes". Errors bars denote the standard error over 5 seeds.

## L. Ablation: Changing the Guidance Weight

In this section, we test if the diversity improvements can be achieved by changing the classifier-free guidance weight. We find that it does improve diversity, however adding our SPELL on top consistently increases the performance further. We use the same SPELL hyperparameters as in the main paper for Latent Diffusion, namely $r = 20$ and overcompensation $1.6$.

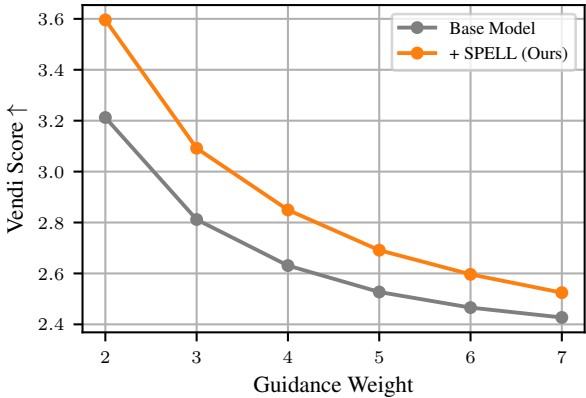

*Figure 19.* Our repellency added to the Latent Diffusion model with different classifier-free guidance weights.

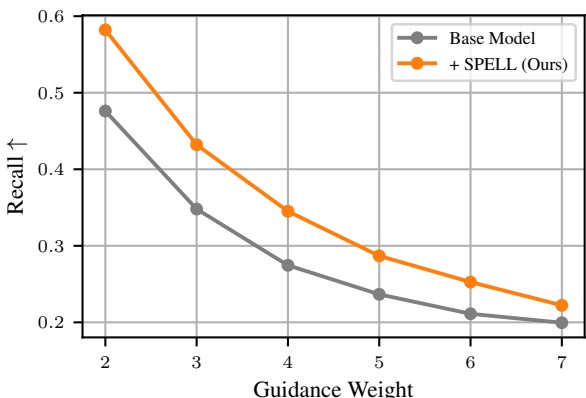

*Figure 20.* Our repellency added to the Latent Diffusion model with different classifier-free guidance weights.

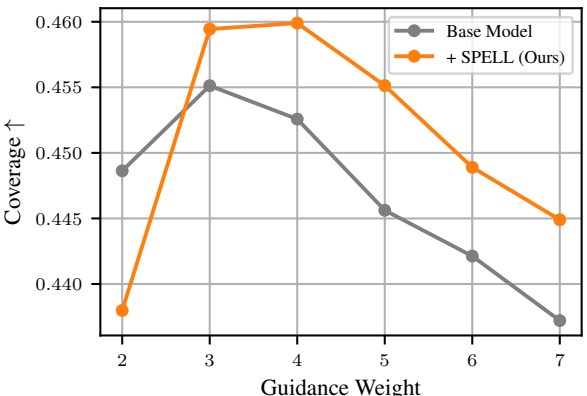

*Figure 21.* Our repellency added to the Latent Diffusion model with different classifier-free guidance weights.

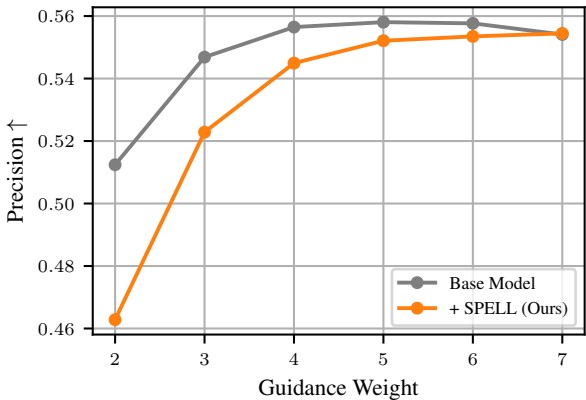

*Figure 22.* Our repellency added to the Latent Diffusion model with different classifier-free guidance weights.

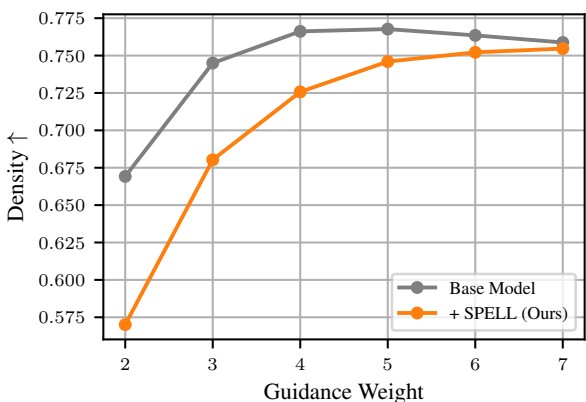

*Figure 23.* Our repellency added to the Latent Diffusion model with different classifier-free guidance weights.

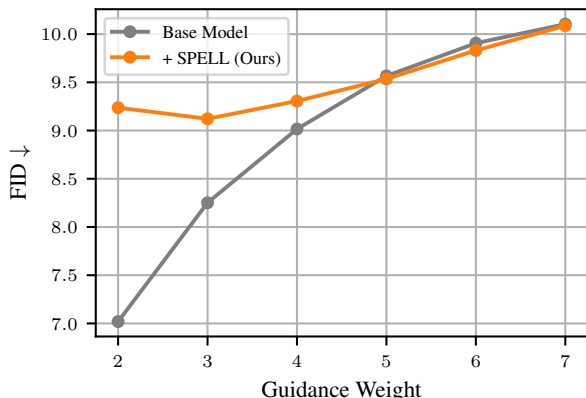

*Figure 24.* Our repellency added to the Latent Diffusion model with different classifier-free guidance weights.

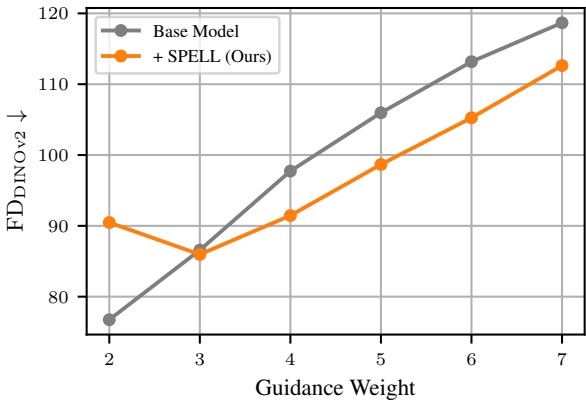

*Figure 25.* Our repellency added to the Latent Diffusion model with different classifier-free guidance weights.

## M. Ablation: Changing the Repellence Radii

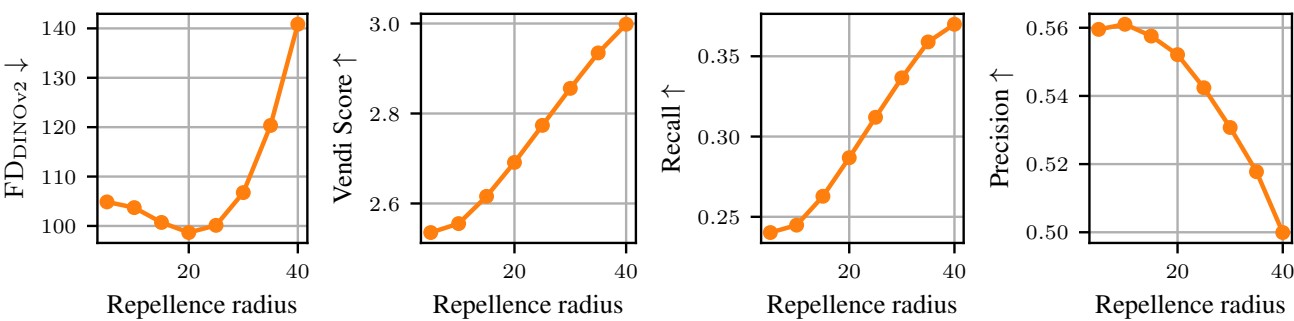

*Figure 26.* Effect of SPELL's hyperparameter $r$ on Latent Diffusion metrics on CC12M. A small radius ($r = 15$) improves the Vendi score, recall, and FD$_{\text{DINOv2}}$ without compromising precision. The radius can be further increased to trade-off precision for additional diversity.

## N. Further Diversity Examples

In order to extend Figure 1, we provide further examples of Simple Diffusion without and with SPELL in Figure 27 to Figure 36. The prompts are chosen from MS COCO, which Simple Diffusion was not trained on. As opposed to Figure 1, this features both of SPELL's capabilities: Intra-batch repellency (every row is a batch of size four), and inter-batch repellency from previous batches, which we treat as the shielded set. The examples affirm qualitatively that SPELL increases the diversity of generated images. Notably, this is without lowering the prompt adherence, which other baselines like IG are prone to, see Table 4 and Figure 4.

We note that some images have copyright overlays, likely learned from the underlying dataset. To investigate this further, we generate 1600 examples for Simple Diffusion without SPELL and 1600 with SPELL. Without SPELL, 62/1600 images have a shutterstock (or similar) overlay, with SPELL it's 105/1600. To confirm that this is a stable trend, we also generate images with Latent Diffusion (which is trained on the same dataset as Simple Diffusion), where it's 79/800 without SPELL and 98/800 with SPELL. While the second result could still be a random chance (Chi-Square independence test with Yates' continuity correction gives p-value = 0.15), the first result is beyond random (p=0.001), and also the effect size is quite measurable (7% vs 4% overlay rate). To improve the understanding of the inner workings, we make two more observations. First, we note that the copyright overlays tend to happen clustered at specific prompts. E.g., one motorcycle prompt has 21/32 images with overlay while most other prompts have 0. So, the distribution is quite skewed and seems to depend on the prompt. Second, we find that the watermarks can serve to push away images from similar ones without the watermark and to pull together images with the same watermark. This informs our best understanding, namely that the copyright overlays serve as a "highway" between modes that allows SPELL to easily explore new modes. SPELL only uses this if the true images / mode distribution of a given prompt actually includes these copyright modes.

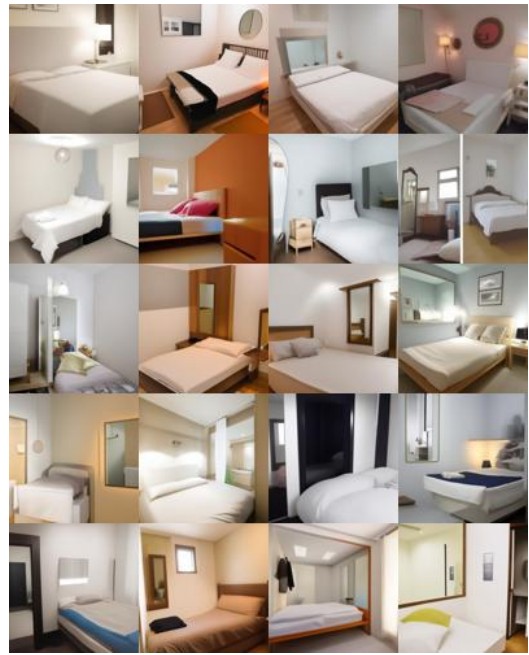 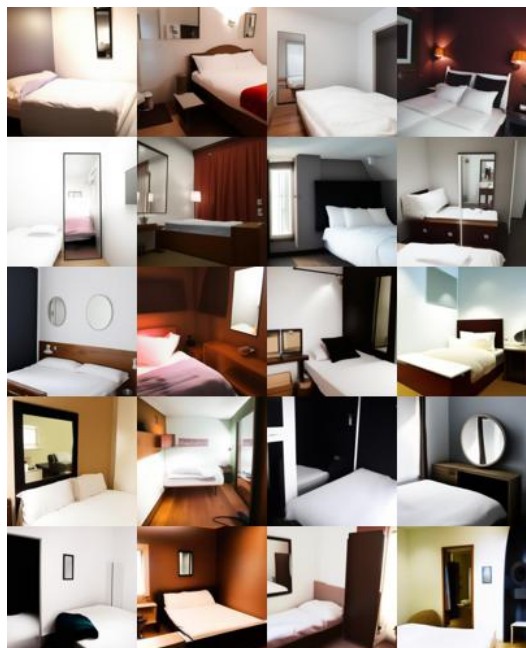

(a) Simple Diffusion without SPELL                    (b) Simple Diffusion + SPELL

*Figure 27.* Images generated with Simple Diffusion without and with SPELL for the MS COCO prompt *"A bed and a mirror in a small room."*. Five batches (rows) with each four images, with both intra- and inter-batch repellency, with the same seeds as the runs without SPELL.

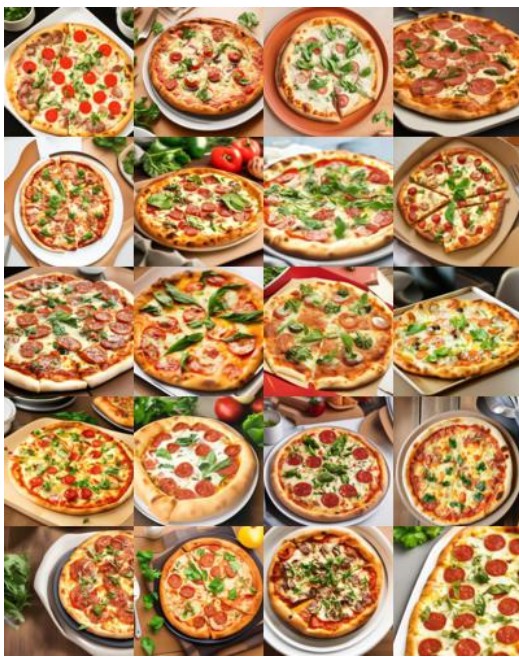 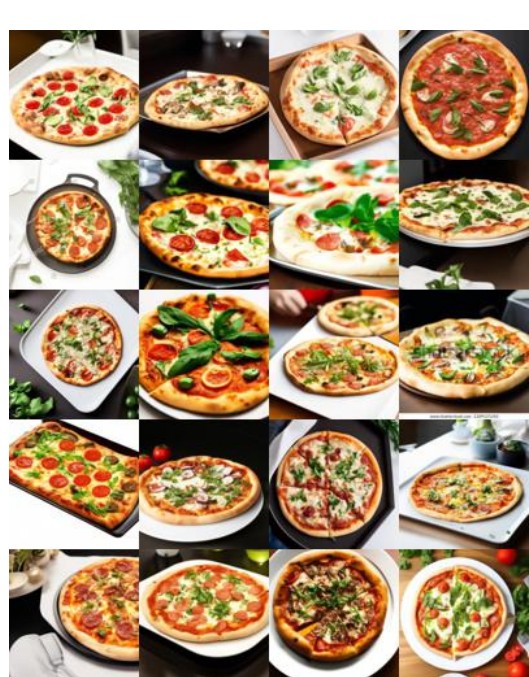

(a) Simple Diffusion without SPELL                    (b) Simple Diffusion + SPELL

*Figure 28.* Images generated with Simple Diffusion without and with SPELL for the MS COCO prompt *"Baked pizza with herbs displayed on serving tray at table."*. Five batches (rows) with each four images, with both intra- and inter-batch repellency, with the same seeds as the runs without SPELL.

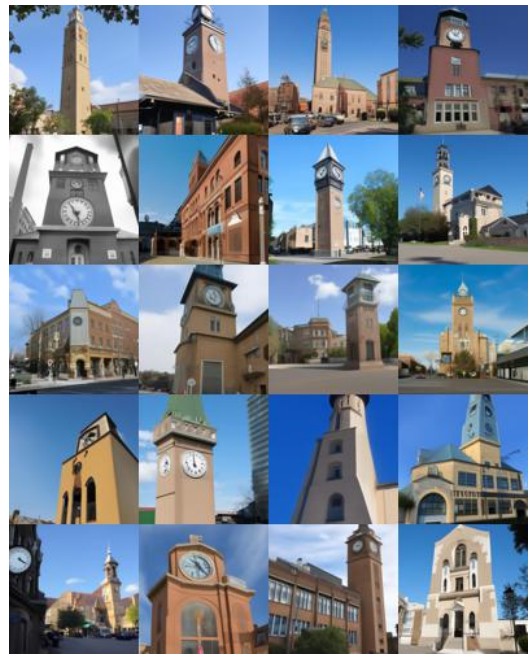 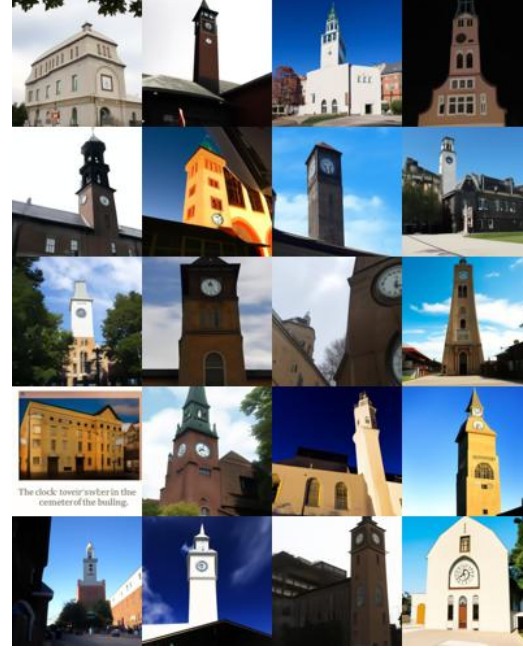

(a) Simple Diffusion without SPELL

(b) Simple Diffusion + SPELL

*Figure 29.* Images generated with Simple Diffusion without and with SPELL for the MS COCO prompt *"The clock tower is in the center of the building."*. Five batches (rows) with each four images, with both intra- and inter-batch repellency, with the same seeds as the runs without SPELL.

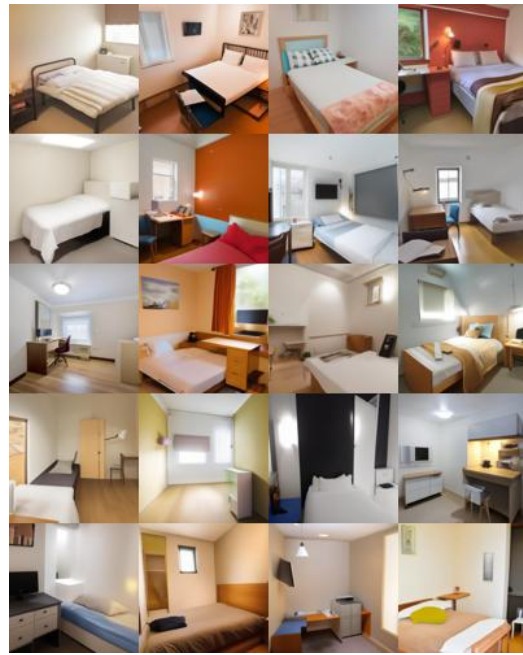 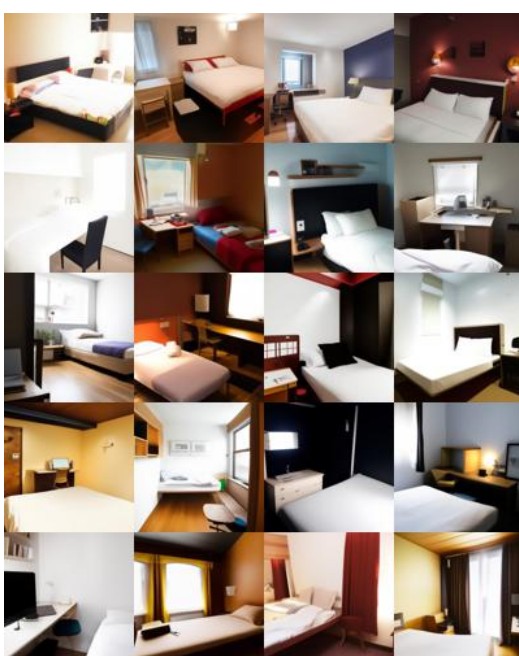

(a) Simple Diffusion without SPELL

(b) Simple Diffusion + SPELL

*Figure 30.* Images generated with Simple Diffusion without and with SPELL for the MS COCO prompt *"A bed and desk in a small room."*. Five batches (rows) with each four images, with both intra- and inter-batch repellency, with the same seeds as the runs without SPELL.

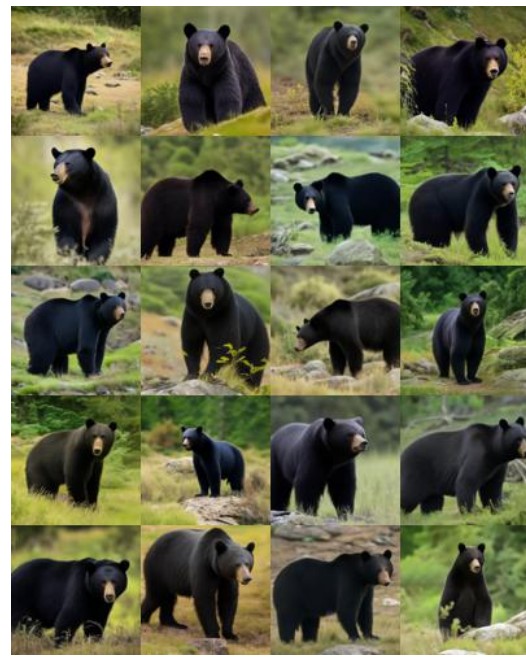 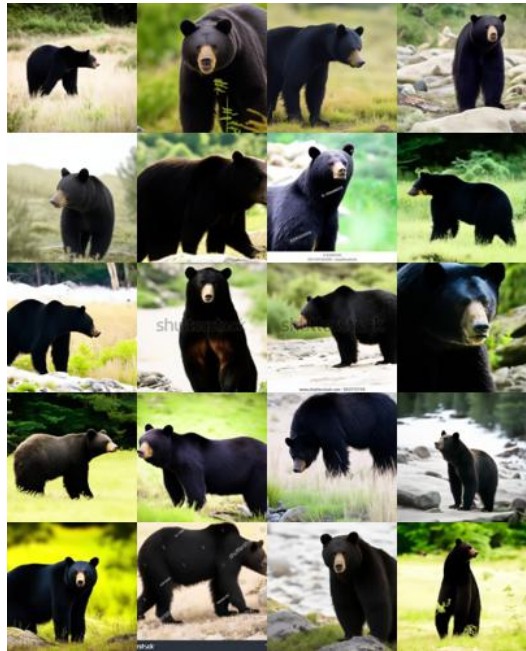

(a) Simple Diffusion without SPELL  (b) Simple Diffusion + SPELL

*Figure 31.* Images generated with Simple Diffusion without and with SPELL for the MS COCO prompt *"A furry, black bear standing in a rocky, weedy, area in the wild."*. Five batches (rows) with each four images, with both intra- and inter-batch repellency, with the same seeds as the runs without SPELL.

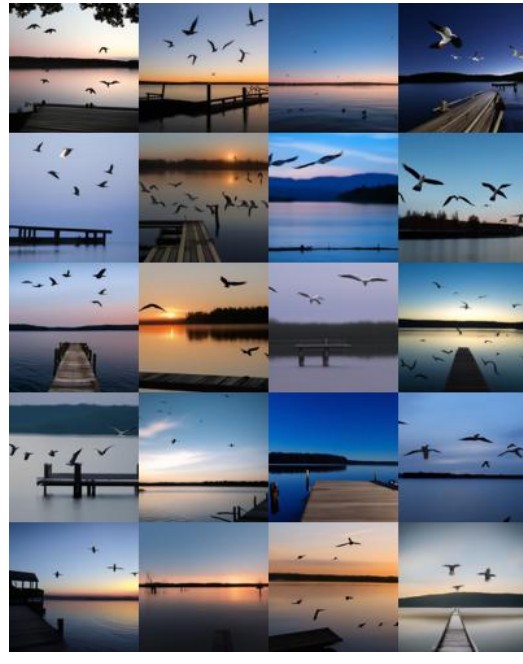 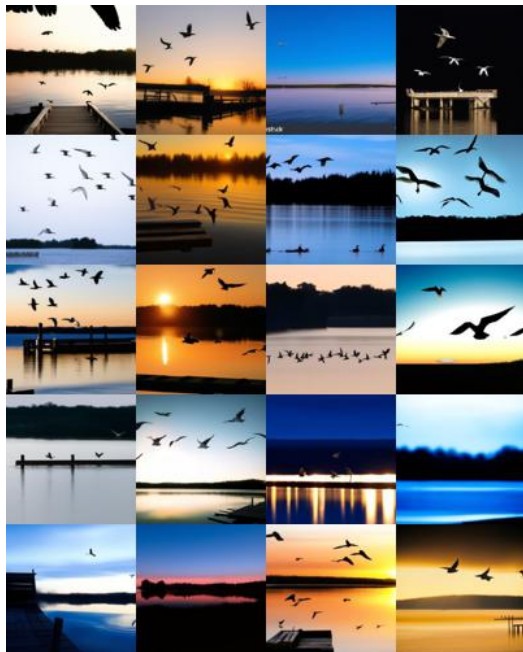

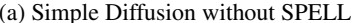

(a) Simple Diffusion without SPELL  (b) Simple Diffusion + SPELL

*Figure 32.* Images generated with Simple Diffusion without and with SPELL for the MS COCO prompt *"A group of seagulls are flying over a wooden dock that is sitting in a lake during the early part of the evening."*. Five batches (rows) with each four images, with both intra- and inter-batch repellency, with the same seeds as the runs without SPELL.

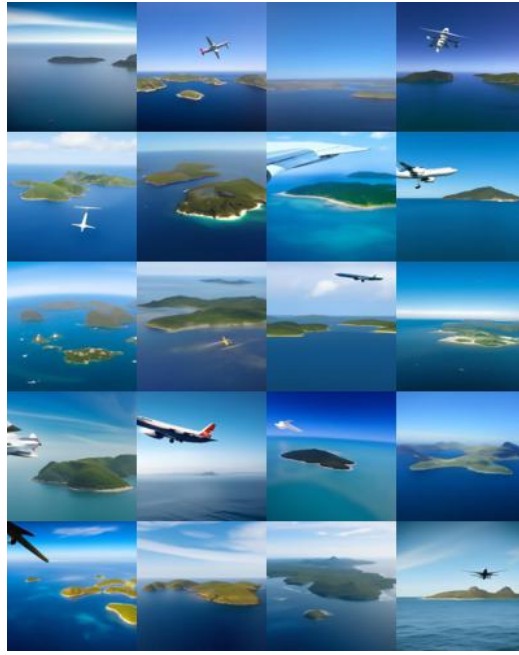 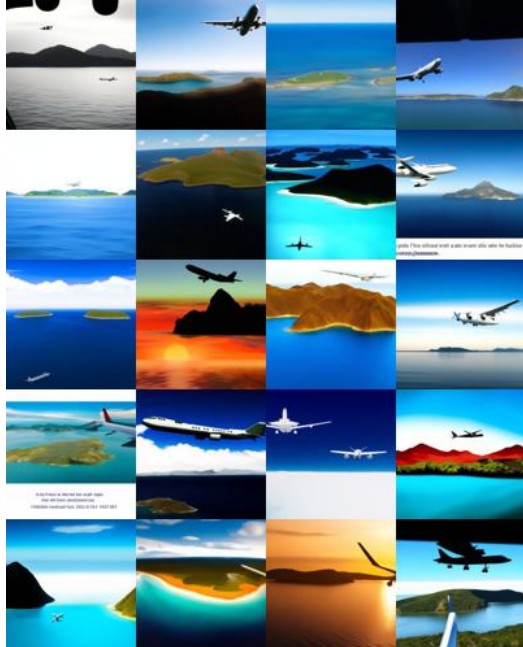

(a) Simple Diffusion without SPELL                    (b) Simple Diffusion + SPELL

*Figure 33.* Images generated with Simple Diffusion without and with SPELL for the MS COCO prompt *"A plane flies over water with two islands nearby."*. Five batches (rows) with each four images, with both intra- and inter-batch repellency, with the same seeds as the runs without SPELL.

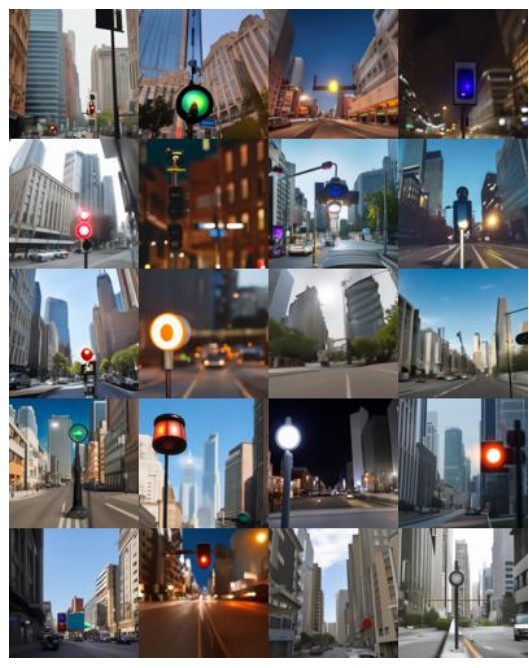 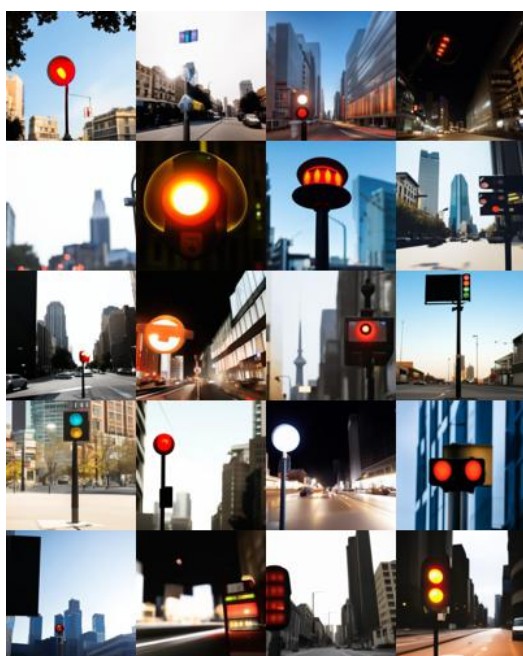

(a) Simple Diffusion without SPELL                    (b) Simple Diffusion + SPELL

*Figure 34.* Images generated with Simple Diffusion without and with SPELL for the MS COCO prompt *"A traffic light over a street surrounded by tall buildings."*. Five batches (rows) with each four images, with both intra- and inter-batch repellency, with the same seeds as the runs without SPELL.

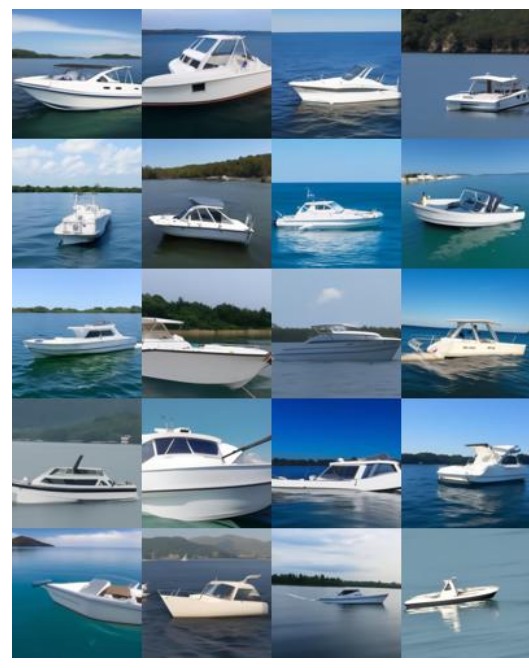

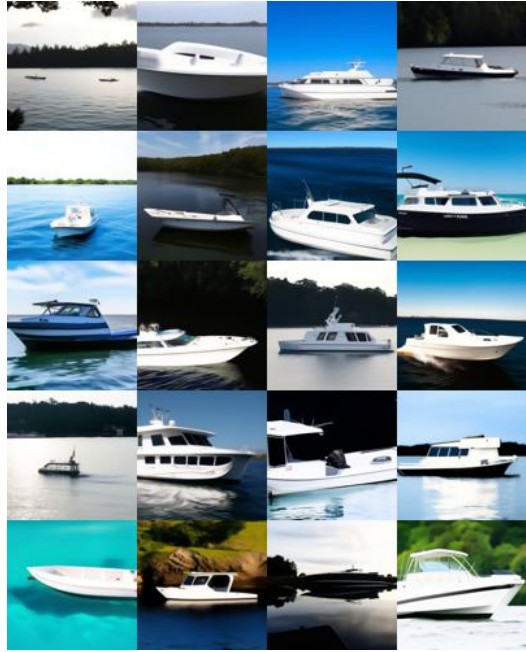

(a) Simple Diffusion without SPELL

(b) Simple Diffusion + SPELL

*Figure 35.* Images generated with Simple Diffusion without and with SPELL for the MS COCO prompt *"a white boat is out on the water"*. Five batches (rows) with each four images, with both intra- and inter-batch repellency, with the same seeds as the runs without SPELL.

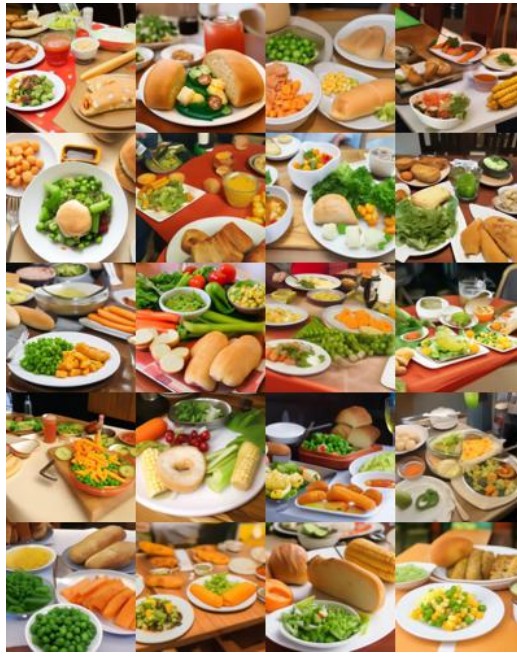

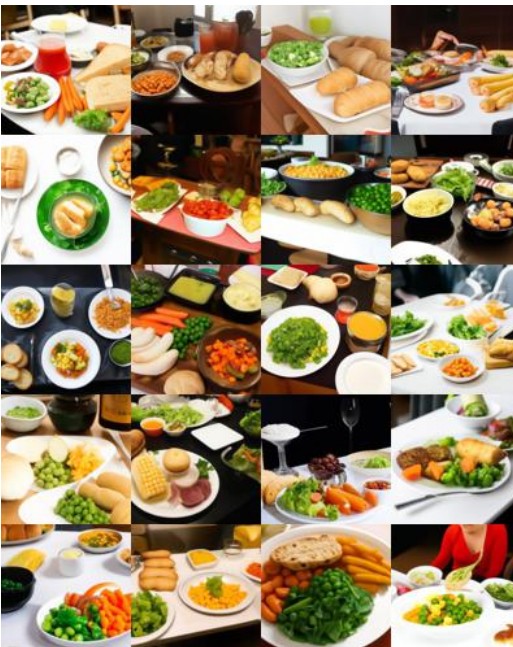

(a) Simple Diffusion without SPELL

(b) Simple Diffusion + SPELL

*Figure 36.* Images generated with Simple Diffusion without and with SPELL for the MS COCO prompt *"A table layed out with food such as, salad, steamed peas and carrots, steamed corn, and bread rolls."*. Five batches (rows) with each four images, with both intra- and inter-batch repellency, with the same seeds as the runs without SPELL.

