# OpenReview forum: "Shielded Diffusion: Generating Novel and Diverse Images using Sparse Repellency"
_ICML.cc/2025/Conference — ICML 2025 poster_

### Official Review · Reviewer_jFPF · 2025-03-12

**Overall Recommendation:** 3

**Summary:**

The paper addresses the issues of limited diversity and replication of training images in text-to-image diffusion models by introducing a method to ensure that generated images are novel and diverse using sparse repellency.

**Claims And Evidence:**

Yes

**Essential References Not Discussed:**

No

**Experimental Designs Or Analyses:**

Yes, the proposed method is validated through comprehensive experiments.

**Methods And Evaluation Criteria:**

Yes

**Other Comments Or Suggestions:**

Some visualization results should be removed to the main context.

**Other Strengths And Weaknesses:**

Strengths:

1. The paper introduces the SPELL method, a novel approach to improve the diversity of generated images in text-to-image diffusion models by using sparse repellency terms.

2.  The proposed SPELL significantly enhances the diversity of generated images without substantially impacting image quality.

3. The paper provides extensive experiment evaluations.

Weaknesses:

1. The SPELL method introduces additional computational overhead due to the repellency terms, potentially making it less efficient than simpler diffusion models.
2. The authors should compare the performance of the proposed SPELL method with varying text prompts, as the diversity of generated images can be significantly influenced by changes in the prompts.
3. The paper lacks organization, making it difficult to follow and identify the key points.

**Questions For Authors:**

I noticed artifacts in Figures 25 and 26, suggesting that the introduced SPELL method may negatively impact the accuracy of the generated images.

**Relation To Broader Scientific Literature:**

The concept of repellency mechanisms has been explored in other areas of machine learning, such as anomaly detection and outlier detection.

**Theoretical Claims:**

No theoretical claims.

---

> ### Author Rebuttal · Authors · 2025-04-01
>
> Thank you for your review and for your assessment that our experimental evaluation is extensive. To add to this, we add the additional experiments you have requested below, namely a runtime analysis and SPELL’s performance under text prompts of varying length. We also provide further results, including qualitative examples, in the responses to the other reviewers.
> ### The SPELL method introduces additional computational overhead due to the repellency terms, potentially making it less efficient than simpler diffusion models.
> A runtime analysis can be found in Table 5 in the appendix, we also post it here for your convenience. It shows that **the generation time with and without SPELL is equal in the diversity setup**. This is because SPELL only adds one matrix operation to calculate distances and one to add the repellency terms, which is small compared to the computational cost of the diffusion backend.
>
> | Model                          | Generation time per image (seconds) |
> |--------------------------------|-------------------------------------|
> | Baseline (Simple Diffusion)    | 2.93 ± 0.12                        |
> | Simple Diffusion + SPELL       | 2.94 ± 0.13                        |
>
> ### The authors should compare the performance of the proposed SPELL method with varying text prompts
> Thank you for the suggestion, **we have added your suggested experiment here** https://imgur.com/a/T6KdRAR . We sliced the CC12M dataset into groups of prompts with increasing length. As can be seen in the plot, SPELL consistently increases the diversity of both short and long prompts compared to the baseline model, without losing on the CLIP-Score, which measures prompt adherence.
> ### The paper lacks organization, making it difficult to follow and identify the key points.
> Thank you for the feedback, we will add intuitions with the additional space of the camera-ready version of the paper.
>
> **Thank you again for your feedback.** We would be happy to learn if the additional experiments and explanations resolve your concerns and update your evaluation of our paper.

---

### Official Review · Reviewer_U4mF · 2025-03-13

**Overall Recommendation:** 3

**Summary:**

This paper introduces an application of negative guidance on reference datasets (e.g., {training, validation} datasets) to enhance diversity and generate novel samples that differ from reference images. The paper leverages geometric steering to guide samples away from the reference dataset. To minimize the performance degradation caused by the proposed method, the smallest perturbation is applied to pull samples outside a ball centered at the datapoint in the reference dataset closest to the current sample. This approach enables the use of diverse images across various forms of latent diffusion, such as text-to-image and image generation. The experimental results demonstrate that the proposed method maintains generation performance while producing more diverse images.

## **Update after rebuttal**

I appreciate the authors graphical explanation of how SPELL performs unconditional generation. While I want to zoom in understanding the limitations and complete failure cases of this paper by varying their hyperparameters and reference datasets, the authors focused on presenting favorable scenarios where SPELL operates smoothly.

Overall, I am satisfied with the current state of this paper because they addressed effective negative guidance, which leads to diverse samples while also generating specific samples away from the reference images. As taking into to all aspects of manuscript and response, I maintain my original score

**Claims And Evidence:**

The paper’s claim is clear and easy to understand. This paper introduces a novel negative guidance approach with a geometric interpretation. This geometric approach seems more clear and straightforward compared to particle guidance and CADS. Empirical evidence shows its superiority over previous methods

However, its limitations lie in its applications to latent diffusion models rather than pixel-space diffusion. This method heavily relies on the hyperparameter $r$, which acts as a reference data shield. However, the suggested value and its analysis are only based on latent diffusion. I believe it needs to validate the proposed method with pixel-space diffusion to enhance its visibility.

**Essential References Not Discussed:**

It would be beneficial to review the paper [1] that discusses negative guidance aiming for mitigation of memorization. Furthermore, [2] also addresses dynamic negative guidance while preserving generation performance and effectively negating a portion of the data distribution.

[1] Chen, Chen, Daochang Liu, and Chang Xu. "Towards memorization-free diffusion models." CVPR2025.
[2] Koulischer, Felix, et al. "Dynamic Negative Guidance of Diffusion Models." ICLR2025

**Experimental Designs Or Analyses:**

Its experimental design evaluates the performance of the model on diversity of generated samples. In particular, the hyperparameter $r$ plays a crucial role in determining both diversity and quality of the generated samples. However, as shown in Table 1, this method does not effectively enhance diversity while preserving image quality. Notably, the FID score is consistently sacrificed in favor of improving recall.

**Methods And Evaluation Criteria:**

This paper compares the proposed model to baselines using appropriate benchmarks and metrics. To assess generating performance, they not only measure {FID}, but also demonstrate {Recall, Coverage, Density} to evaluate the diversity of the generated images by the proposed method.

**Other Comments Or Suggestions:**

None

**Other Strengths And Weaknesses:**

### **Weakness**

- This paper demonstrates that the proposed method consistently improves recall metrics, but the FID score is not maintained. However, the paper does not provide an explanation for this issue. While the FID score also considers the diversity of generated samples, the numerical evidence presented in Table 1 suggests that the generated images lack plausibility. I believe that this phenomenon is asked for further investigation.

**Questions For Authors:**

None

**Relation To Broader Scientific Literature:**

This paper suggests that negative guidance can be interpreted geometrically.

**Theoretical Claims:**

This paper doesn’t rigorously offer a theoretical framework for negative guidance using diffusion models. However, their conditions and the magnitude of negative guidance are based on a geometric interpretation. Their messages are understandable in terms of measuring distances and determining how close the current sample is to reference data points.

---

> ### Author Rebuttal · Authors · 2025-04-01
>
> Thank you for your review and for your interest in the theory behind SPELL that enables its outperformance. We delve into the theoretical connections between SPELL and other methods below, as well as the experimental results for pixel-space diffusion. We also provide further results in the responses to the other reviewers.
> ### I believe the paper needs to validate the proposed method with pixel-space diffusion to enhance its visibility.
> We already validate on a pixel-space diffusion model, Simple Diffusion, as noted in lines 239 and 292. Please refer to Table 1 for its results. In general, **SPELL works independently of the space the diffusion model diffuses in**, whether it is a latent VAE space or a direct RGB diffusion, and whether it is an unguided, classifier-guided, or classifier-free guided model. This sets it apart from alternative approaches like CADS and IG.
> ### This paper doesn’t rigorously offer a theoretical framework for negative guidance using diffusion models. However, their conditions and the magnitude of negative guidance are based on a geometric interpretation.
> SPELL’s repellency directions and magnitudes are based on a theoretical framework, which is explained in appendices A to C. The general problem we tackle is that we want to bring the probability / density that a diffusion model generates (an r-Ball around) a point $z_k$ to exactly zero, i.e. $P_0(B_r(z_k)) = 0$. The interesting part about this proof is that we need to choose the direction and magnitude of the intervention despite not knowing the density at $P_0(z_k)$ of the original diffusion model, since diffusion models do not provide density estimates. In Appendix B **we use Tweedie’s formula for a theoretical derivation of the exact direction and magnitude required to achieve this, leading to Equation (5)**. While in the main paper, we start with a geometric interpretation of Equation (5), we also give a conservative-field interpretation in Appendix C to connect SPELL to other negative-guidance based frameworks, and show SPELL’s theoretical relation to Diffusion Posterior Sampling in L165-219. We hope that these rigorous theoretical analyses help future researchers connect and develop SPELL as well as negative guidance.
> ### The proposed method consistently improves recall metrics, but the FID score is not maintained. However, the paper does not provide an explanation for this issue.
> We attribute the decrease in FID to the fact that it is a reference-based metric that compares distributional similarity with a reference dataset. As SPELL increases the diversity and explores more of the image manifold, its output distribution is intentionally broader than the reference dataset that the diffusion models without SPELL are trained to match as closely as possible. This issue occurs with all reference-based metrics (precision, coverage, recall, density), which is why we report reference-free metrics like the Vendi and CLIP score. **SPELL’s trade-off is, however, better than for other diversity-inducing metrics** (CADS, IG, PG in Fig. 3), even for reference-based metrics. __To demonstrate the preservation of image quality, we’ve generated additional high-resolution SD3 + SPELL images here__ https://imgur.com/a/Dsh7gxb https://imgur.com/a/Uz0LhOz https://imgur.com/a/VGcvqNE https://imgur.com/a/zrZRuV2 https://imgur.com/a/cuSAADZ .
> ### It would be beneficial to review the paper [1] that discusses negative guidance aiming for mitigation of memorization. Furthermore, [2] also addresses dynamic negative guidance while preserving generation performance and effectively negating a portion of the data distribution.
> Thank you for those interesting pointers! We will discuss them in the camera-ready version of the paper (this year's reviewing form does not allow to upload revised pdfs).
>
> **Thank you again for your time during the busy rebuttal period.** We would be happy to learn if the additional experiments and explanations resolve your concerns and update your evaluation of our paper.

---

> > ### Comment · Reviewer_U4mF · 2025-04-01
> >
> > Thanks for clarifying my initial review. However, I have a question that remains unclear after the author’s rebuttals.
> >
> > ### Experiments on primitive diffusion models without conditional information
> >
> > All experimental situations strongly favor conditional information, such as “text prompts” and “class labels.” I kindly request that “SPELL” be integrated with simple generation models, either “EDMv1” or “Improved DDPM,” without conditional information. Otherwise, simple examples like (Two-moon, Star, etc.) cases demonstrate this approach more effectively; The modality is not a concern; (The authors don’t need to validate this experiment on high-resolution images).
> >
> > In addition to scientific writing, presenting limitations on the proposed method is crucial for contributing to academic fields. Based on my experience, conditional generation can guide a sampling trajectory more easily than vanilla diffusion models.
> > I believe that demonstration in various forms of diffusions models and presentation of failure cases or limitations strengthen this paper effectively.
> >
> > ——— Post authors response ———
> >
> > I appreciate the authors graphical explanation of how SPELL performs unconditional generation. While I want to zoom in understanding the limitations and complete failure cases of this paper by varying their hyperparameters and reference datasets, the authors focused on presenting favorable scenarios where SPELL operates smoothly.
> >
> > Overall, I am satisfied with the current state of this paper because they addressed effective negative guidance, which leads to diverse samples while also generating specific samples away from the reference images. As taking into to all aspects of manuscript and response, I maintain my original score

---

> > > ### Author Response · Authors · 2025-04-03
> > >
> > > Thank you for your fast response. We are happy to provide the additional experiment below.
> > >
> > > ### Two-moons experiment for unconditional generation
> > >
> > > Sure, SPELL can also be used on unconditional diffusion models, because it does not rely on the conditioning signal (other than CADS and IG). **We provide your requested two-moons example here** https://imgur.com/a/JlRi8uR . The two sets of samples are generated with the exact same noise seeds, once without and once with SPELL. It shows that SPELL has a blue-noise like pattern that covers the distribution better than non-diversified sampling. We will add this toy experiment to the camera-ready version to better demonstrate how SPELL behaves. The reason we have used text-to-image and class-to-image models in the paper so far is that they are the most popular ones and present an interesting challenge where we need to become diverse without losing prompt/class adherence (hence the CLIP score tradeoff curves).
> > >
> > > ### Limitations section
> > >
> > > Thank you for the suggestion. We agree that openly discussing limitations strengthens the contribution, which is why we discuss limitations, e.g., in L39-40 (abstract), L308-312 (left), L406-411 (right), 427-433 (left), L425-439 (right). With the additional page of the camera-ready version, we will introduce a dedicated limitations section to bundle these and motivate future research avenues.
> > >
> > > **Thank you again for your time and fast interaction.** We would be happy to learn if these and the previous additions update your evaluation of our paper.

---

### Official Review · Reviewer_eLTh · 2025-03-14

**Overall Recommendation:** 2

**Summary:**

This paper introduces Shielded Diffusion, which aims to generate images outside of protected sets. These protected sets may include protected images, other data in the current batch, or data used during training. The authors determine whether the diffusion trajectory is expected to fall into protected sets during the sampling process, dynamically triggering a repellency term to ensure the sampling endpoint stays away from protected sets. The proposed method is training-free, capable of enhancing generation diversity while reducing the risk of model infringement (preventing the generation of training data).

## update after rebuttal
Although the author's reply to some extent resolved my concerns, I still believe that there is some room for improvement in the writing of this paper, as reviewer jFPF also mentioned "The paper lacks organization, making it difficult to follow and identify the key points."
In addition, in the author's reply, I found that flux+spell (https://imgur.com/a/FwJvkhG) seems not to work very well, with a relatively small difference before and after, and it affects the image quality (It seems to have affected the image harmony, such as color temperature, saturation.).

**Claims And Evidence:**

This paper claims that the proposed technique enhances the generation diversity of diffusion models while only slightly disturbing the FID. It compares text-to-image and class-conditional diffusion models in Table 1, compares with other diversity-inducing methods in Figure 3, and conducts further Sparsity Analysis and qualitative result comparisons, which to some extent validate the paper's claim.

**Essential References Not Discussed:**

Based on my understanding of this field, the authors appear to have included most of the relevant literature discussions.

**Experimental Designs Or Analyses:**

This paper conducts experiments on class-to-image and text-to-image diffusion models, measuring multiple metrics including Recall, Cendi, FID, and Coverage. Additionally, it performs Sparsity Analysis, with experimental design and analysis that are reasonable to some extent.

However, I believe it could provide examples and analysis under the SD3 model with higher resolution images and more complex prompts, as practical usage tends to favor the generation of complex scenes and high-resolution images.

**Methods And Evaluation Criteria:**

The method proposed in this paper helps with copyright protection and improves generation diversity, which is meaningful in practical applications.

**Other Comments Or Suggestions:**

In the introduction, the authors mention "This phenomenon is illustrated in Figure 1 for three popular diffusion models, Stable Diffusion 3 (Esser et al., 2024), Simple Diffusion (Hoogeboom et al., 2023) and MDTv2 (Gao et al., 2023)." However, Figure 1 only shows results for Simple Diffusion and MDTv2, with no results for Stable Diffusion 3.

**Other Strengths And Weaknesses:**

Strengths: The method proposed in this paper helps with copyright protection, which is meaningful for mitigating risks in AI-generated content; the proposed method is training-free and does not introduce additional computational overhead.

Weaknesses: This paper is obscure and difficult to understand, especially for non-domain experts. The reading experience is particularly poor in terms of logical flow. I believe the authors should express their content using more accessible, standardized, and logically structured narratives. For example, Sparse Repellency is a core concept of this paper, and the authors should explain this term first.

**Questions For Authors:**

1. Could the authors explain why they did not conduct some experiments on larger resolution images (greater than 256*256)?
2. Could the authors explain why they did not test more complex scenarios beyond "a dog plays with a ball," and whether more diverse generation could be achieved in such cases?
3. Is there additional computational overhead introduced, and could they provide an analysis of this overhead?
4. Could the authors distill the paper's greatest contribution, rather than listing 5 lengthy points as in the Introduction?

I will maintain a rejection stance; further explanation from the authors would help me improve my score.

**Relation To Broader Scientific Literature:**

Sparsity is the biggest difference between this paper and other methods [1][2][3] controlling similar diversity-precision trade-offs. Other methods apply disturbances to change diffusion trajectories throughout the entire sampling time steps, while the method proposed in this paper uses ReLU weighting for dynamic activation, adding no disturbance when trajectories are moving toward more diverse directions. This helps ensure quality while improving diversity.

[1] Kynkäänniemi T, Aittala M, Karras T, et al. Applying guidance in a limited interval improves sample and distribution quality in diffusion models[J]. arXiv preprint arXiv:2404.07724, 2024.
[2] Sadat S, Buhmann J, Bradley D, et al. CADS: Unleashing the diversity of diffusion models through condition-annealed sampling[J]. arXiv preprint arXiv:2310.17347, 2023.
[3] Corso G, Xu Y, De Bortoli V, et al. Particle guidance: non-iid diverse sampling with diffusion models[J]. arXiv preprint arXiv:2310.13102, 2023.

**Theoretical Claims:**

This paper's geometric explanation for SPELL is reasonable, as it aims to add disturbance to push $\hat{x}_0$ away when it potentially falls within radius r (equations 4 and 5).

---

> ### Author Rebuttal · Authors · 2025-04-01
>
> Thank you for your review and for acknowledging that SPELL tackles meaningful practical applications while remaining training-free. We are happy to provide the experimental results for high-resolution images, complex prompts, and compute overhead that you have requested below.
> ### Examples and analysis under the SD3 model with higher resolution images and more complex prompts
> **We have added 1024x1024 images with prompts of varying length, generated both via SD3 and FLUX**, under the following links: https://imgur.com/a/Dsh7gxb https://imgur.com/a/Uz0LhOz https://imgur.com/a/VGcvqNE https://imgur.com/a/zrZRuV2 https://imgur.com/a/cuSAADZ https://imgur.com/a/oHouASK https://imgur.com/a/3mI2ynU https://imgur.com/a/knqPFhc https://imgur.com/a/FwJvkhG https://imgur.com/a/qigDza4 . We’ve generated the images one-by-one from the same seeds like in Figure 5. The results show that while the first samples of the unconditional model are already diverse enough (top rows) and SPELL does not act due to its sparsity, it starts diversifying the images in the later generations (bottom rows). This holds both for simple and complex prompts.
>
> The experiments in our paper also already use complex prompts, with the CC12M prompts ranging from 15 to 491 characters. **We have added an ablation on SPELL’s diversity for different prompt lengths** here https://imgur.com/a/T6KdRAR . It shows that SPELL increases the diversity of the generated images even in the longest 90-100% percentiles of the prompts compared to the baseline model, without compromising on the prompt adherence measured by the CLIP score.
> ### I believe the authors should express their content using more accessible, standardized, and logically structured narratives.
> Thank you for the feedback! We will highlight the narrative in the camera-ready version (this year's reviewing form does not allow to upload revised pdfs).
> ### Figure 1 only shows results for Simple Diffusion and MDTv2, with no results for Stable Diffusion 3.
> We apologize for this editing issue. **We provide a Stable Diffusion 3 example here**, https://imgur.com/a/qNYX5tX , on top of the other SD3 examples linked above.
> ### Experiments on larger resolution images (greater than 256*256)
> We have added results for 1024x1024 images, see the links above.
> ### More complex scenarios beyond "a dog plays with a ball,". Can more diverse generations could be achieved in such cases?
> The “a dog plays with a ball” prompt was only used in Figure 5 to give a qualitative example. **Our quantitative experiments already use CC12M prompts with 15-491 characters, and we have added qualitative results for longer prompts in the links above**, following your recommendation. We will outline this in the camera-ready version.
> ### Is there additional computational overhead introduced, and could they provide an analysis of this overhead?
> We provide an analysis of computational overhead in Table 5 in the Appendix for the diversity use-case, and also reposted here. As can be seen, **SPELL does not increase the generation time during diverse generation**, since it adds only one matrix operation to calculate distances and one to calculate the repellency direction (and in many timesteps does not even do this thanks to its sparsity (see Appendix H)) which is a small computation compared to the diffusion backbone.
>
> | Model                          | Generation time per image (seconds) |
> |--------------------------------|-------------------------------------|
> | Baseline (Simple Diffusion)    | 2.93 ± 0.12                        |
> | Simple Diffusion + SPELL       | 2.94 ± 0.13                        |
> ### Could the authors distill the paper's greatest contribution
> We provide SPELL, a sparse and training-free mechanism to push diffusion trajectories away from already-generated or protected images. We will make this clearer in the abstract and introduction, thank you for your feedback!
>
> **Thank you again for your reviewing time.** We would be happy to learn if the additional experiments and explanations resolve your concerns and update your evaluation of our paper.

---

### Official Review · Reviewer_E31r · 2025-03-17

**Overall Recommendation:** 3

**Summary:**

This paper proposes sparse repellency (SPELL) to prevent diffusion models from generating images in a set of L2 balls. SPELL can be used to (1) protect diffusion models from generating training images; (2) encourage diversity between multiple generations. Extensive experiments shows the superiority of SPELL.

**Claims And Evidence:**

Claim #1: SPELL is a new effective method for preventing diffusion models from generating samples inside shields (which are essentially L2 ball neighborhoods of points in a reference set).

Evidence: The methodology and discussion in Section 3 explains the SPELL method and its relationship to related works very well.

Claim #2: SPELL is empirically effective in terms of shielded generation of diffusion models.

Evidence: The experimental results in Section 4 shows SPELL can improve diversity with a little drop in image quality. In Section 4.3, the authors compare SPELL with other methods and presents a better Pareto front. In Section 4.6, SPELL decreases the proportion of samples   inside shields for EDMv2 in ImageNet-1k, demonstrating its potential in large-scale setting and data privacy protection.

**Essential References Not Discussed:**

N/A

**Experimental Designs Or Analyses:**

1. Why does SD3 + SPELL (row2 and row 5 in Table 1) has significant degradation in FID and FD_DINO? Does this imply drop in image quality?

2. Can image protection with SPELL be applied to text-to-image models? Compared with the method in Section 4.6, T2I may need retrieval augmented generation to apply SPELL. Could the authors comment on this?

**Methods And Evaluation Criteria:**

The method and evaluation criteria looks reasonable to me.

**Other Comments Or Suggestions:**

N/A

**Other Strengths And Weaknesses:**

Overall, I think the submission proposes a simple, reasonable method that is suitable for two important problems in sampling diffusion models, training image protection and diversity enhancement. The authors also did extensive experiments to support their method.

**Questions For Authors:**

1. which model does Figure 3 compare? There are 3 Text-to-Image models listed in the setting part, SD3, Latent Diffusion and Simple Diffusion.

2. Can the SPELL method be applied to rectified flow-based models? For example, FLUX.

3. The assumption of $B_k$ being disjoint in L#165-183, page 4, limits the range of $r$ ( if $r \rightarrow \infty$, $B_k$ will inevitably overlap). Can the authors provide a principal way for choosing $r$ for a specific reference set?

**Relation To Broader Scientific Literature:**

The paper proposes a new method for training image protection and diversity enhancement for diffusion model inference. I regard these two problems as important problems for diffusion models. The proposed method, SPELL, although being simple, is a sensible method for these two methods that may enlighten future research on this direction.

**Theoretical Claims:**

I didn't redo the proof by myself, but the connection between SPELL and DPG/PG makes sense.

---

> ### Author Rebuttal · Authors · 2025-04-01
>
> Thank you for your review and for recognizing that SPELL is a simple way to tackle two current problems in diffusion models at once. We are happy to share explanations and your requested FLUX experiments below.
> ### Can image protection with SPELL be applied to text-to-image models?
> Yes, there is no difference in applying SPELL for image protection on class-to-image, noise-to-image, or text-to-image diffusion models. Unlike approaches like Interval Guidance and CADS, *SPELL is generalized away from the conditioning signal*, i.e., it does not rely on classifier- or classifier-free guidance, but acts on the diffusion trajectory and the output space of any diffusion model itself, even if the conditioning signal tries to push the diffusion trajectory close to a protected image. **SPELL can thus protect any-to-image diffusion models**. The reason why we demonstrate this on the class-to-image model MDTv2 in Section 4.6 as opposed to a text-to-image model is simply that MDTv2’s exact training dataset and preprocessing are public (ImageNet-1k) so that we can protect the exact training data, whereas text-to-image models like SD3 are trained on proprietary dataset splits.
> ### Why does SD3 + SPELL have significant degradation in FID and FD_DINO? Does this imply a drop in image quality?
> We attribute the decrease in FID to the fact that it is a reference-based metric that compares distributional similarity with a reference dataset. As SPELL increases the diversity and explores more of the image manifold, its output distribution is intentionally broader than the reference dataset that SD3 without SPELL is trained to match as closely as possible. This issue occurs with all reference-based metrics (precision, coverage, recall, density), which is why we report reference-free metrics like the Vendi and CLIP score. **SPELL’s trade-off is, however, better than for other diversity-inducing metrics** (CADS, IG, PG in Fig. 3), even in reference-based metrics. **We’ve also generated additional high-resolution SD3 + SPELL images**, generated one-by-one with the same seeds as in Figure 5, to show the image quality https://imgur.com/a/Dsh7gxb https://imgur.com/a/Uz0LhOz https://imgur.com/a/VGcvqNE https://imgur.com/a/zrZRuV2 https://imgur.com/a/cuSAADZ
> ### Can the SPELL method be applied to rectified flow-based models? For example, FLUX.
> Yes, SPELL can be applied to any diffusion model. **We have added examples for FLUX1.schnell** in 1024x1024: https://imgur.com/a/oHouASK https://imgur.com/a/3mI2ynU https://imgur.com/a/knqPFhc https://imgur.com/a/FwJvkhG https://imgur.com/a/qigDza4  Since FLUX1 already generates more diverse images by itself compared to SD3, SPELL’s sparsity means that it is activated on fewer pictures, and in more details, like background objects or colors. If you are interested in one-step diffusion models, we denote that SPELL should be used with a number of diffusion steps greater than 1, because it is applied after each diffusion step. If there is only one step, it will be applied effectively after the generation has already ended, which could introduce artifacts. We will add this discussion to the camera-ready version of the paper to outline SPELL’s applicability.
> ### Which model does Figure 3 compare?
> Figure 3 shows results for Latent Diffusion, as denoted in the caption.
> ### The assumption of $B_k$ being disjoint in L165-183, page 4, limits the range of $r$ ( if $r \rightarrow \infty$, $B_k$ will inevitably overlap). Can the authors provide a principal way for choosing $r$?
> In the protection case, the question of how big of a shield a user wants to create around a protected image is ultimately a user choice. Should they use a radius that large that shields start overlapping, and should a trajectory point exactly to the middle of two shields (and inside the radius of both shields), their terms cancel each other out, as we openly address in L179-180. In this case, the overlapping shields $z_1$ and $z_2$ can be merged with a combined radius of $r_\text{merged} = r + d(z_1, z_2)$ and $\frac{z_1+ z_2}{2}$ as shield center to retain the protection guarantee, as mentioned in L673 in the appendix. However, in practice we did not observe this to be necessary – even in the protection experiment on all 1.2M ImageNet-1k images, we did not have issues with overlapping shields. Due to the high-dimensional space, it is unlikely that a trajectory points exactly in the middle of two shields, where their terms would cancel each other out exactly. Rather, their terms point in slightly different directions, so that SPELL guides the trajectory points away from both shields throughout the diffusion. So, **a user is able to choose a shield radius as large as it suits their needs**. As a side note, for the diversity use case, we provide a principled way of choosing $r$ in L779 in the appendix.
>
> **Thank you again.** We would be happy to learn if the additional experiments and explanations update your evaluation of our paper.

---

### Decision · Program_Chairs · 2025-05-01

**Decision:**

Accept (poster)

**Comment:**

The paper received 3 Weak Accept and 1 Weak Reject scores pre-rebuttal. The rebuttal did not affect the reviewers' decisions. Overall, the reviewers applauded the paper for its novel and reasonable method that effectively addressed training image protection and diversity enhancement, as well as the extensive experiment evaluations. Reviewer eLTh still concerned about paper writing and unsatisfactory image quality produced by FLUX+SPELL, just keeping his Weak Reject score unchanged.

The ACs checked and found the concern from Reviewer eLTh minor. The contributions of the paper outweigh its shortcomings. Hence, we agree to accept the paper to ICML.